# New Definitions and Evaluations for Saliency Methods: Staying Intrinsic and Sound

## Abstract

Saliency methods seek to provide human-interpretable explanations for the output of machine learning model on a given input. A plethora of saliency methods exist, as well as an extensive literature on their justifications/criticisms/evaluations. This paper focuses on heat maps based saliency methods that often provide explanations that look best to humans. It tries to introduce methods and evaluations for masked-based saliency methods that are *intrinsic* — use just the training dataset and the trained net, and do not use separately trained nets, distractor distributions, human evaluations or annotations. Since a mask can be seen as a "certificate" justifying the net's answer, we introduce notions of *completeness* and *soundness* (the latter being the new contribution) motivated by logical proof systems. These notions allow a new evaluation of saliency methods, that experimentally provides a novel and stronger justification for several heuristic tricks in the field (T.V. regularization, upscaling).

## 1 Introduction

*Why did the deep net give a certain answer on a particular input, and can we trust the answer? Saliency methods* try to provide explanations, and are thus of great interest from viewpoint of human explainability, fairness, robustness, etc. This paper restricts attention to methods that return an importance score for each coordinate of the input —usually visualized as a heat map— that captures its importance to the final decision.[1] We refer the reader to Samek et al. (2019) for an extensive survey and Section 2 for a short account of such methods and controversies.

There are two important components in the research on saliency: *saliency methods* that produce such heat maps for explanations, and *saliency evaluation metrics* that aim to test and compare saliency methods. Numerous saliency methods have been proposed to learn such heat maps. Some methods learn maps through "credit attribution" to individual input coordinates by using methods reminiscent of backpropagation (Binder et al., 2016; Selvaraju et al., 2019), while some are derived using careful axiomatization of credit assignment using the idea of Shapley values from cooperative game theory (Lundberg & Lee, 2017b; Yeh et al., 2020). Some recent methods train another deep net to produce heat maps (Dabkowski & Gal, 2017; Phang et al., 2020). Given the proliferation of saliency methods, an ecosystem of evaluation metrics has emerged to evaluate the quality of explanations produced by saliency methods, either through human evaluations (Adebayo et al., 2018), comparison to certain ground truth explanations (Zhang et al., 2018) or other evaluations that do not require any external annotation or supervision (Dabkowski & Gal, 2017) Petsiuk et al. (2018).

In this paper we restrict attention to a class of evaluations we refer to as *intrinsic*, that aim to evaluate saliency maps based on whether they are good explanations for the model prediction. These only involve computations using the provided heat map and the net, and do not involve extrinsic factors such as human judgements or retraining. A popular idea in such evaluations (see Section 3 for references) is to create a new composite input —or sequence of such inputs— using the heat map and the original input, and to evaluate the original net on this composite input. For example, if $M$ is a binary vector with 1's in the $k$ coordinates with the highest values in the heat map, then $x \odot M$ (with $\odot$ denoting coordinate-wise multiplication) can be viewed as a composite input[2] (aka

---

[1] Heat maps suffice for recognition/classification tasks; other tasks may require more complex explanations.
[2] When $x$ is an image the zeroes in this masked input are often replaced with gray values.

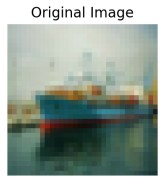 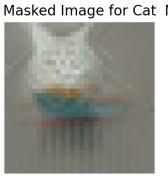 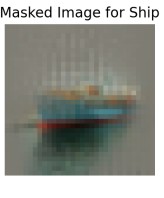 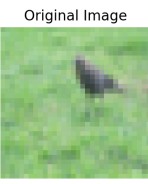 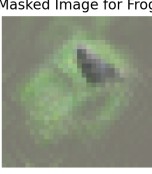 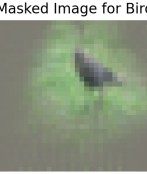

Original Image | Masked Image for Cat | Masked Image for Ship | Original Image | Masked Image for Frog | Masked Image for Bird

Figure 1: Masked CIFAR-10 images generated by our procedure with $\lambda_{TV} = 0.01$ shows the artifacts exist for masks generated for incorrect labels. More examples can be find in Figure 7 in Appendix. The base classifier outputs the correct label on the original image (ship and bird resp.) with probability at least 0.99, and assigns probability at most $10^{-5}$ for the incorrect label (cat and frog resp.). With the generated masks, the AUC metric for the correct label remains high (around 0.94 and 0.90), which corresponds to *completeness*, but AUC metric for the incorrect label rises tremendously (around 0.18 for cat mask, 0.71 for frog mask.) This suggests violation of *soundness*.

"masked input") that can be fed into the original net. By varying $k$ one obtains a sequence of masks $M$. Although there is an issue of distribution shift due to the net never having trained on composite inputs, in practice, especially for image data, trained nets work fine on masked inputs. (This is also exploited in Shapley value based saliency methods.)

Inspired by logical reasoning, we identity a key component missing in existing *intrinsic evaluations*: *soundness* of the saliency method. In simple terms, checking soundness for mask-based explanations entails verifying that it is impossible for the method to produce well-performing (in the composite-input evaluations) heat maps corresponding to the *incorrect* labels, i.e. labels not predicted as the most likely by the model. By contrast current masking evaluations focus on a property akin to *completeness*[3] in logic: the heat map "justifies" the correct label, i.e. the label judged to be most likely by the model. We argue that soundness can be a useful criterion in addition to completeness.

The concept of soundness is motivated by observing that the recent mask-based saliency methods are inherently different from the earlier gradient-based methods. They produce a heatmap that is interpreted as a masked image that, when fed into the net must make it output the same label that received the top logit value on the full image. Our paper observes that it is unclear at a logical level why this should be considered an "explanation", since to be convinced one must verify that a different mask could not induce the net to output a label corresponding to other logits. Figure 1 shows that in fact this issue is real; masks do exist that can "justify" the wrong labels. We introduce soundness as a quantitative measure for this issue (definitions in Section 3.1), and propose that evaluations of mask-based saliency methods should account for soundness. Many past works allude to difficulty of the soundness concept, mentioning that computations to find the "best" mask end up finding artifacts that can even justify labels that are obviously incorrect. To avoid uncovering artifacts, the methods to find masks/heat maps use a suitable regularization (usually TV). However, Figure 1 illustrates that TV regularization is not a full solution to removing artifacts.

Another reason for incorporating soundness with completeness in metrics is to implicitly encourage saliency methods to also produce saliency maps for incorrect labels. When using the classifier in the wild—as opposed to on a dataset with centered images— we might encounter multiple salient objects in the image, and the net may not have high confidence in a single label. It would then be reasonable to have saliency methods find evidence for all the labels which the net outputs with reasonable confidence. We find in Figure 2 that our saliency method that produces masks for all labels can produce different and meaningful masks for different labels when there are multiple objects present in the image[4], compared to prior state-of-the-art methods, that output a single mask for all labels that contains both objects (more on this in Appendix B).

---

[3]In the literature devoted to saliency methods based upon backpropagation-like procedure, such as GRADIENT ⊙ INPUT or LRP, the term *completeness* is used in a slightly different sense: the sum of the heat map values has to be equal to the logit value for the label.

[4]In this paper, multiple objects in one image is not formally considered, as we assume single object classification is the prior knowledge. Therefore, either zebra or elephant in Figure 2 is regarded as an incorrect label. A quick extension to the multi-object case is to define correct labels as the labels whose model probability exceeds a certain threshold. Even without the extended definition of correct labels, Definition 3.2 with suitable $\alpha, \beta$ already allows explanations to exist for both elephant and zebra as long as the model probability for elephant and zebra are high enough.

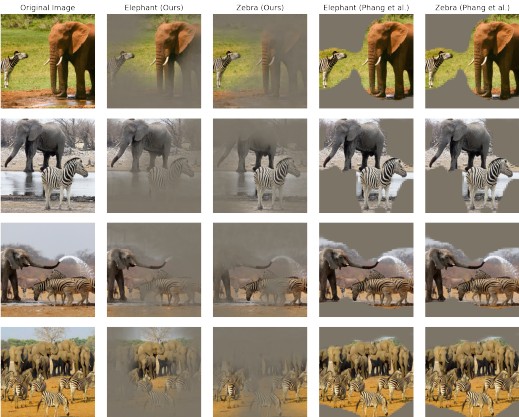

Figure 2: Images containing both elephant(s) and zebra(s), and the corresponding masked ones generated by our method and the best-performing CA model in Phang et al. (2020). The masks by Phang et al. (2020) are identical for different labels, and contains both elephant and zebra. In contrast, our method outputs descent masks for elephant and zebra accordingly.

**Main contributions:** In this paper we make a case for pushing saliency methods and metrics to incorporate soundness in addition to completeness. We propose metrics and a method and experimentally evaluate them in comparison to previous methods and metrics, leading to some novel findings. Our main contributions are as follows:

- A new single number metric named *consistency score* that evaluates the completeness and soundness simultaneously. It measures the probability that the saliency map for the model prediction has the best quality compared to all other labels. See Section 3.2 for a formal definition.

- A saliency method in Section 4 that bears similarity to prior works that learns masks through optimization, but with subtle differences. One of the differences is based on the logic of soundness: we learn a mask by trying to maximize the probability for the given label, rather than matching the probability, and learn the *best mask* for all labels this way. We find in experiments (Section 6) that our method not only performs comparably to prior masked-based methods on earlier saliency metrics, but also does better than those methods on the proposed consistency score.

- Previous works proposed "tricks" like TV penalty and upscaling of masks (Fong & Vedaldi, 2017; Dabkowski & Gal, 2017) that are motivated by their ability to avoid "artifacts" and make the saliency maps visually look better. These justifications, however, are not in the spirit of an intrinsic evaluation of saliency maps. In this paper, we provide a novel *intrinsic* justification for these choices of TV penalty and upscaling, by showing that they improve the saliency maps by making them more sound, as empirically demonstrated in Section 6. We complement this finding with a theoretical result in Section 5 that proves that TV regularization can help with soundness in a simple linear classification setting.

**Paper outline:** Section 2 discusses prior works on saliency methods and evaluations. Section 3 delves further into the concept of completeness and soundness in the context of saliency evaluation, and describes our proposed metric of consistency score. Section 4 describes our saliency method and how it differs from prior approaches. Finally Section 5 describes various supportive experiments.

## 2 PRIOR APPROACHES

We delegate a more thorough description of prior work to Appendix D, but mention some common saliency and saliency evaluation methods here. Saliency methods aim to explain a model's decision about an input. Saliency evaluation methods aim to evaluate the goodness of a saliency method.

**Saliency methods** include *backpropagation based approaches* such as Gradient $\odot$ Input (Shrikumar et al., 2017), LRP (Binder et al., 2016), GradCAM (Selvaraju et al., 2019), Smooth-Grad (Smilkov et al., 2017). Another line of work is *masking methods* which include techniques based on averaging over randomly sampled masks (Petsiuk et al., 2018), optimizing over meaningful mask perturbations (Fong & Vedaldi, 2017), and real time image saliency using a masking network (Dabkowski & Gal, 2017). Pixels that have been removed from the image by the mask may be replaced by greying out, by Gaussian blurring as in (Fong & Vedaldi, 2017), or with infillers such as CA-GAN (Yu et al.,

2018) used in (Chang et al., 2018; Phang et al., 2020), or DFNet (Hong et al., 2019). De Cao et al. (2020) find masks using differentiable masking. *Boolean logic* is another approach for saliency methods (Ignatiev et al., 2019b;a; Macdonald et al., 2019; Mu & Andreas, 2020; Zhou et al., 2018).

**Arguments about saliency.** As saliency methods have arisen, discussion about them has occurred e.g. in (Seo et al., 2018; Fryer et al., 2021; Gu et al., 2018; Sundararajan & Najmi, 2020). Some works reveal situations where Shapley axioms work against feature selection or where Shapley values may be calculated in conflicting ways (Fryer et al., 2021; Sundararajan & Najmi, 2020). Others question why noise adding saliency methods work (Seo et al., 2018), with some pointing out that some explanations are not class discriminative (Gu et al., 2018).

**Saliency evaluation methods.** Extrinsic evaluation metrics include the **WSOL** metric, and **Pointing Game** metric proposed by Zhang et al. (2018) and ROAR (Hooker et al., 2019). Other more intrinsic methods include early saliency evaluation techniques like **MorF** and **LerF** (Samek et al., 2016), **Insertion and Deletion game** proposed by Petsiuk et al. (2018), which involve either inserting pixels in order of most importance or deleting pixels in order of most importance. **BAM** (Yang & Kim, 2019) creates saliency maps by pasting object pixels from MSCOCO (Lin et al., 2014) The **Saliency Metric** proposed by Dabkowski & Gal (2017) thresholds saliency values above some $\alpha$ chosen on a holdout set, finds the smallest bounding box containing these pixels, upsamples and measures the ratio of bounding box area to model accuracy on the cropped image, $s(a, p) = \log(\max(a, 0.05)) - \log(p)$ where a is the area of the bounding box and p is the class probability of the upsampled image.

**Controversies.** Brunke et al. (2020) show that perturbation methods are sensitive to baseline and Petsiuk et al. (2018) point out that human centric explanations (based on bounding boxes) may not reveal why the model made a certain decision. Our notion of intrinsic saliency method aligns with the latter idea, and we introduce a new criteria for evaluating saliency maps: soundness, which captures the notion that maps should not produce explanations for low probability labels.

## 3 MASKING EXPLANATIONS AND COMPLETENESS/SOUNDNESS

A running theme in saliency methods/evaluations is that the salient pixels should be sufficient to convince us about the model's output, regardless of contents of the other pixels. For instance, greying out (or setting them to average pixel value) non-salient pixels should have very little effect on the output. This idea appears in many saliency methods (Dabkowski & Gal, 2017; Phang et al., 2020) (and evaluation metrics (Petsiuk et al., 2018)), including Shapley values and mask-based explanations. This motives the following definition of a masked-based explanation.

**Definition 3.1** (Masking Explanation)**.** A masking explanation for input $x$ is a distribution $\Delta$ over subsets $S$ of input coordinates ("salient sets of $x$") as well as an input modification process $\Gamma$ that generates a distribution of *modified inputs* $\tilde{x} \sim \Gamma(x, S)$ under the constraint that $\tilde{x}$ matches $x$ on every coordinate in set $S$. We use $\Gamma(x, \Delta)$ to denote the distribution of $\tilde{x}$ produced by $\Gamma$ when the input is $x$ and the set $S$ is sampled from $\Delta$.

A salient set $S$ could be the direct output of any saliency method, or the output heat map passed through a potentially randomized discretization (e.g., the insertion game in Section 3.3), leading to a distribution $\Delta$ over sets $S$. Simple examples of the input modification process $\Gamma$ that stay in the *intrinsic* framework are: greying out the pixels outside $S$, replacing them by pixels from a Gaussian blurring of $x$ (Fong & Vedaldi, 2017). Another example of $\Gamma$ —albeit non-intrinsic and hence not used in this paper— uses a conditional image generative model to produce new pixel values in $\overline{S}$ conditional on pixels in $S$ being consistent with $x$ (Agarwal & Nguyen, 2020; Chang et al., 2018). Definition 3.1 can also be changed to allow $\Gamma$ to change the values of pixels in $S$. In our proposed method in Section 4, we consider another distribution $\Gamma$, which replaces pixels outside of $S$ with pixels from a random image from the training set[5]. This amounts to grafting salient pixels of $x$ on top of a random image, reminiscent of BAM evaluations (Yang & Kim, 2019) for saliency methods.

### 3.1 COMPLETENESS AND SOUNDNESS

We now describe completeness and soundness for saliency evaluations. In a multiclass classification setting, denote by $f(x, a)$ the probability for label $a$ returned by the model on input $x$. The prediction of the model on input $x$ is $\hat{y}(x) := \arg\max_{a \in C} f(x, a)$, where $C$ denotes all classes. For $p \in [0, 1]$,

---

[5]We think other random image distributions should work too.

a masking explanation $(\Delta, \Gamma)$ for input $x$ is said to *p-validate* the label $a \in C$ if

$$\mathbb{E}_{\tilde{x} \sim \Gamma(x, \Delta)}[f(\tilde{x}, a)] \geq p. \tag{1}$$

We say that a masking explanation $x$ $p$-validates the model prediction if it $p$-validates $\hat{y}(x)$.

**Definition 3.2** (Completeness and Soundness). Fix an input modification process $\Gamma$. A saliency method given $f, x$ and any label $a \in C$ produces a distribution $\Delta(x, a)$ over salient sets. For $\alpha, \beta \leq 1$, the method is $\alpha$-*complete on* $f, x, a$ if the masking explanation $(\Delta(x, a), \Gamma)$ $p$-validates the label $a$ for $x$ with $p = \alpha \cdot f(x, a)$, and the method is $\beta$-*sound on* $f, x, a$ if the masking explanation $(\Delta(x, a), \Gamma)$ cannot $p$-validate the label $a$ with $p > \frac{1}{\beta} \max\{f(x, a), \epsilon\}^6$.

Intuitively, when $\alpha, \beta$ are close to 1, $\alpha$-completeness means that if the model outputs a high probability for label $a$, then the probability for label $a$ after seeing only the coordinates in the salient sets is also high; $\beta$-soundness means that if the model outputs a low probability for label $a$, then the probability for label $a$ after seeing only the coordinates in the salient sets is also low. Thus a desirable saliency method should have $\alpha$-completeness for labels predicted by the model and $\beta$-completeness for other labels. This captures the idea that we want the method to be able to validate the model prediction $\hat{y}(x)$ but not other labels. It is worth noting that the saliency method that declares all pixels to be salient is $\alpha$-complete and $\beta$-sound with $\alpha = \beta = 1$, but its size is too large to provide any useful information. Thus one should ask the saliency method to search over the salient sets of smaller sizes. But artifacts for incorrect labels may occur at small size (Figure 1), which hurts soundness. For this reason we put explicit or implicit constraints on the salient sets (e.g., TV regularization) and ask saliency methods to be $\alpha$-complete and $\beta$-sound with $\alpha, \beta$ as close to 1 as possible. Ideally we should adjust the constraints to achieve a good tradeoff between completeness and soundness. In our experiments, we find that standard "tricks" like TV regularization and mask upscaling provide such good tradeoffs, as observed in Table 3 and Figure 3.

**Important note.** In the logic setting, from where we borrow the concept of soundness, one has to search among all possible proofs to ensure that there is indeed no valid proof for any false proposition. Similarly, our soundness metric is non-vacuous only if the saliency method "makes its best efforts" to find masking explanations for every label. Otherwise the saliency method could for instance ignore the value of $a$ return the mask for $\hat{y}$ which can $p$-validate $\hat{y}(x)$ with a high value of $p$ and thus not validate any other label, leading to a false notion of soundness. If the saliency method behaves like this on the incorrect label (i.e., is not working at producing an explanation) then as a backup the evaluation can use some default method to produce saliency explanations for those labels. Our method for mask-based explanations, described later, is simple and not tied to any specific philosophy or a-priori definition of saliency. It can produce explanations for all labels.

## 3.2 CONSISTENCY SCORE FOR EVALUATING COMPLETENESS AND SOUNDNESS

Inspired by the notions of completeness and soundness, we propose to use a single number metric that evaluates the completeness and soundness simultaneously over a data distribution.

**Definition 3.3** (Consistency Score). Let $\mathcal{X}$ be the distribution of input $x$ and $\Gamma$ be an input modification process. For a saliency method that produces salient sets $\Delta(x, a)$ on input $x$ and label $a$, we define the "saliency score" for $(x, a)$ as $g_\Delta(x, a) = \mathbb{E}_{\tilde{x} \sim \Gamma(x, \Delta(x, a))}[f(\tilde{x}, a)]$, i.e. average probability the model assigns to a labels for modified versions of $x$ using the label-specific salient sets $\Delta(x, a)$. The consistency score for the saliency method is defined as the probability that the saliency score is consistent with the model prediction, i.e.,

$$\Pr_{x \sim \mathcal{X}} \left[ \arg\max_a g_\Delta(x, a) = \hat{y}(x) \right]. \tag{2}$$

It is easy to see that if a saliency method is 1-complete and 1-sound on all inputs, then the consistency score is 1. The following lemma shows that even $\alpha$-completeness and $\beta$-soundness can imply consistency if there is a gap between the probabilities for the largest and second largest labels.

**Lemma 3.4.** *For an input $x$, if a saliency method is $\alpha$-complete on the model prediction $\hat{y}(x)$ and $\beta$-sound on all other labels, and if the ratio between the output probabilities for the largest label $\hat{y}(x)$ and second largest labels $\frac{f(x, \hat{y}(x))}{\max\{\max_{a \neq \hat{y}(x)}\{f(x, a)\}, \epsilon\}}$ is larger than $\frac{1}{\alpha\beta}$, then the saliency method is consistent with the model on input $x$.*

---

[6]The constant $\epsilon$ is used to avoid blowing up due to tiny probabilities.

*Proof.* By $\alpha$-completeness, the saliency method $p$-validates $\hat{y}(x)$ with $p = \alpha \cdot f(x, \hat{y}(x))$. Since $\frac{f(x,\hat{y}(x))}{\max\{\max_{a \neq \hat{y}(x)}\{f(x,a)\}, \epsilon\}} > \frac{1}{\alpha\beta}$, $p$ has the lower bound $p > \frac{1}{\beta}\max\{\max_{a \neq \hat{y}(x)}\{f(x,a)\}, \epsilon\}$ for all $a \neq \hat{y}(x)$. Then $\beta$-soundness implies that the saliency method cannot $p$-validate $a$. Therefore $\mathbb{E}_{\tilde{x} \sim \Gamma(x, \Delta(x,a))}[f(\tilde{x}, a)]$ is maximized when $a = \hat{y}(x)$, and hence consistency. $\square$

### 3.3 CONNECTION TO POPULAR AREA-UNDER-THE-CURVE EVALUATIONS OF HEAT MAPS

As mentioned, our approach requires the method to output salient sets whereas existing methods often return a heatmap of saliency values. However, popular Area-Under-the-Curve (AUC) evaluation metrics (Petsiuk et al., 2018) of saliency methods can be reinterpreted in terms of our completeness and soundness. An example is the *insertion game*:

**AUC of Insertion game:** *For $s = 1$ to $d$ take the top $s$ saliency values, and plot the probability given by model to label $a$ on the input where the top $s$ pixels of $x$ are retained and remaining pixels are assigned values from the input modification process $\Gamma$. Return AUC.*

**Lemma 3.5.** *Given a saliency heatmap with AUC $\rho$ for label $a$, we can convert it into a masking explanation that $\rho$-validates the label $a$.*

*Proof.* Given a heatmap, produce the salient set $S$ by picking $s$ uniformly at random from 1 to $d$ and letting $S$ be coordinates corresponding to top $s$ saliency values. Let $\Delta$ be the distribution of the produced salient sets. Then $\mathbb{E}_{\tilde{x} \sim \Gamma(x, \Delta)}[f(\tilde{x}, a)] = \rho$ by definition. $\square$

## 4 PROCEDURES TO FIND MASKING EXPLANATIONS

Based on the definition 3.1 of masking explanations, we propose a very simple method to find saliency masks with good empirical completeness and soundness scores. We introduce our methods in this section concisely. The detailed intuitions and implementations are provided in Appendix A.

Our method bears similarity to prior work on learning mask based saliency maps, but with subtle differences. The key difference from Dabkowski & Gal (2017); Phang et al. (2020) is that we do not use a neural network to learn the mask and the difference from Fong & Vedaldi (2017); Agarwal & Nguyen (2020); Chang et al. (2018) is that we use a different input modification process $\Gamma$ (see Definition 3.1) to infill other pixels during training: given $x$ and $S$, generate a hybrid inputs whereby the pixels in the set $S$ match with $x$ and the pixels outside $S$ are set to those of a random image drawn from the training set $\mathcal{X}$. Replacing with random image pixels instead of gray pixels will make it harder to find a mask. Despite this, we can still learn a mask that predicts the correct label with high confidence in practice. Furthermore, it achieves superior performance over replacing with gray pixels, which is likely because the added hardness of task helps increase the robustness of the output. The final crucial difference is that our method can find a mask for every input-label pair $(x, a)$ and does so by trying to maximize the probability assigned to label $a$ on modified images, as opposed to trying to match the model probability for $a$.

As is standard, we relax the domain of masks $M$ from binary $\{0,1\}^{hw}$ to continuous $[0,1]^{hw}$, and optimize $M$ for input $x$ and label $a$ based on the following natural objective[7]

$$L(M; (x,a)) = \mathbb{E}_{\bar{x} \sim \mathcal{X}}\left[-\log(f(M \odot x + (1-M) \odot \bar{x}, a))\right] + \lambda_1 \|M\|_1, \quad (3)$$

where the part of $x$ on $M$ is superimposed onto a distractor $\bar{x} \sim \mathcal{X}$ as $\tilde{x} = M \odot x + (1-M) \odot \bar{x}$, and the $\ell_1$ norm penalty on $M$ helps to reduce the size of masks.

We further employ Total-Variation (TV) penalty (Fong & Vedaldi, 2017) and upscaling of the mask from a lower resolution one (Petsiuk et al., 2018) by learning a low-resolution mask at scale $s$, $M \in [0,1]^{hw/s^2}$, to minimize the following

$$L(M; (x,a)) = \mathbb{E}_{\bar{x} \sim \mathcal{X}}\left[-\log(f(M^{\times s} \odot x + (1-M^{\times s}) \odot \bar{x}, a))\right] + \lambda_{TV}TV(M^{\times s}) + \lambda_1\|M^{\times s}\|_1$$
$$(4)$$

where $M^{\times s} \in \mathbb{R}^{hw}$ is obtained by upscaling $M$ by a factor of $s \in \{1, 4\}$ via bilinear interpolation.

While the motivation cited for these two "tricks" is to avoid artifacts, it is not clear whether artifacts are a bad thing, since they might be relevant to the net's decision-making. Indeed, in our experiments, we show that while TV penalty or upscaling does produce better looking masks, they lead

---

[7]A standard way to maximize probability is to minimize the negative log probability

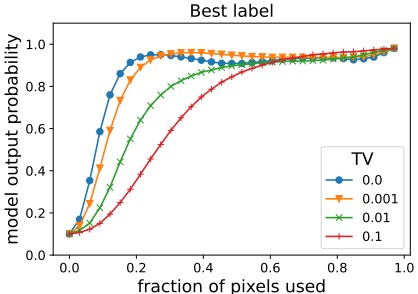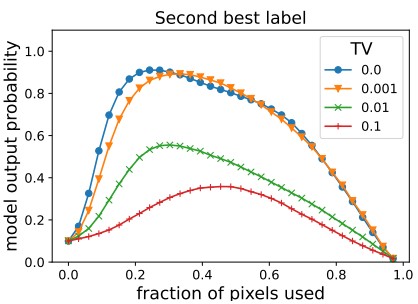

Figure 3: Plot of model output probability as more pixels from the original image are retained using learned masks. The remaining pixels are replaced with gray. Different curves correspond to different values of TV regularization ($\lambda_{TV}$). Larger area-under-curve (AUC) for the left figure (best label) suggests good completeness, while lower AUC for the right figure suggests good soundness. Plots suggest that adding TV significantly helps with soundness, while only slightly hurting completeness.

to a drop in the *completeness metric*. However we also show that adding such tricks leads to significant improvement in the *soundness metric,* thus providing a novel justification for the use of such tricks, beyond just the heuristic argument of getting rid of artifacts. In Section 5 we also provide theoretical justification for why TV penalty can help with soundness, even for the simple case of linear predictors on non-image data.

## 5 CLARIFYING BENEFIT OF TV REGULARIZATION

Our experiments show that the main benefit of TV regularization in saliency methods is that it improves soundness. Here we sketch a simple example showing how this benefits even for simple linear models on non-image data. Details appear in Appendix E. Consider $\mathcal{S}$, a dataset of labeled data $(\boldsymbol{x}, y)$ where labels $y$ are binary, and a linear classifier with weight $\boldsymbol{w}$ and margin $\gamma > 0$. Assume inputs are of unit norm and with bounded $\ell_\infty$ norm and so is $\boldsymbol{w}$. The saliency method has to return a single subset $S$ of salient coordinates to validate the label. The input modification process $\Gamma$ will assign 0 to all coordinates outside $S$. Question is whether the decision can be validated using a small $S$. *A priori* this seems impossible. Furthermore imposing a TV constraint seems to help nothing, because the solution has no obvious continuity structure.

But now assume we randomly permute the coordinates. This (paradoxically) turns out to make a big difference! For simplicity we consider salient sets among intervals, in other words sets of TV $\leq 2$.

**Theorem 5.1.** *For any $(\boldsymbol{x}, y) \in \mathcal{S}$, after random shuffle following holds for any $L_1 = \Omega(\frac{1}{\gamma^2} \log \frac{1}{\delta})$, $L_2 = \Omega(\frac{1}{\gamma^2} \log \frac{d}{\delta})$, where probability is over the random shuffle:*

  *1. (Completeness) With probability $1 - \delta$, there is an interval of length $L_1$ that validates $y$;*

  *2. (Soundness) With probability $1 - \delta$, no interval of length $L_2$ can validate $-y$.*

## 6 EXPERIMENTS

In this section, we present results for our mask learning procedures described in Section 4 and our consistency score metric described in Definition 3.3. We also analyze the role of TV penalty and upsampling. We observe from the results that (1) our simple mask learning procedure achieves competitive performance with existing saliency methods on existing metrics; (2) our procedure does better on our consistency score metric compared to previous methods, which suggests that consistency score has no major conflicts with previous metrics and that our method is more sound; (3) TV regularization and upscaling significantly aid soundness, while only slightly hurting completeness. Overall, consistency score increases with higher TV regularization and upscaling, thus showing the ability of our metrics to provide *intrinsic explanations* for existing "tricks" that previously only had extrinsic justifications.

Experiments involving consistency score are computationally infeasible to perform on ImageNet as it contains 1000 classes. Therefore, these experiments are performed either on CIFAR-10 or Imagenette (Howard, 2020), a 10-class ImageNet subset. Even for CIFAR-10 and Imagenette, since

Table 1: We compute AUC of insertion game with gray infilling as masking explanation. The last two columns are the mask-model consistency conditioned on whether the base classifier predicts the ground truth $y$. The best two of each column are marked bold. From the table, our method with upscaling factor $s = 4$ achieves the best consistency.

| | Deletion ↓ | Insertion (gray) ↑ | Saliency Metric ↓ | Consistency Score ↑ | Consistency when $\hat{y} = y$ ↑ | Consistency when $\hat{y} \neq y$ ↑ |
|---|---|---|---|---|---|---|
| Gradient ⊙ Input | **0.42** | 0.56 | 0.31 | 0.841 | 0.977 | 0.27 |
| Real time saliency | 0.48 | 0.66 | **−0.85** | 0.857 | 0.980 | 0.34 |
| Fong & Vedaldi (2017) | 0.64 | 0.63 | −0.41 | 0.845 | **0.990** | 0.23 |
| Phang et al. (2020) | **0.43** | **0.76** | −0.27 | **0.871** | **0.985** | 0.39 |
| Ours ($s = 1$) | 0.52 | **0.68** | **−0.91** | 0.862 | 0.968 | **0.41** |
| Ours ($s = 4$) | 0.47 | 0.57 | −0.66 | **0.880** | 0.980 | **0.46** |
| Random | 0.45 | 0.45 | −0.35 | 0.829 | 0.964 | 0.26 |

consistency score requires computing 10 times more saliency maps than other metrics, we test it on 1000 randomly drawn images from original test set. We also run experiments that do not involving consistency score on various datasets (including ImageNet and CIFAR-100) with various models. Results are shown in Appendix C. Details of training procedure (beyond those in Section 4) are also in the Appendix C.

## 6.1 FULL COMPARISON TO EXISTING METRICS AND METHODS ON IMAGENETTE

We compare our procedure to other methods including Real Time Saliency (Dabkowski & Gal, 2017), Gradient ⊙ Input (Shrikumar et al., 2017), Fong & Vedaldi (2017) and Phang et al. (2020) on Imagenette. Apart from our new consistency score metric, we calculate the Deletion Game and Insertion Game metrics using the code provided by `https://github.com/eclique/RISE`, and Saliency metric (SM), an another intrinsic evaluation metric from Dabkowski & Gal (2017). Detailed description of the different metrics can be found in the Appendix.

For our method, we use the learned mask $M$ from optimizing the objective function in Equation (4) with $\lambda_{TV} = 0.01$ and $s = \{1, 4\}$ as the saliency map. For fairness, we use the identical ResNet50 pretrained on ImageNet as the base classifier for every saliency method. The salient sets used in our consistency score follows the Insertion game. Insertion scores were calculated by replacing remaining pixels with gray. We normalize the maps so that all values lie in $[0, 1]$ before use.

To get a sense what the numbers mean, we also include the performance of a random mask, where the score for each pixel is sampled independently from standard normal distribution. The results are shown in Table 1. For the deletion metric, we note that most methods have comparable or worse performance than the random mask, which suggests that the metric does not give us much signal about the goodness of the saliency maps. For the Insertion scores and saliency metric, our method performs decently since these metrics mostly depend on variety of the completeness condition.

For our consistency score metric, most previous methods perform well on consistency score when the base classifier predicts the ground truth. However, their performance on samples that the base classifier predicts incorrectly is not as good as ours. Our method shows that soundness can be improved without much affecting performance on other known metrics. Note that our metric assumes that testing methods take effort to generate different masks for each possible label. Saliency methods like Phang et al. (2020) actually inherent advantage on our metric by not doing that. Considering that our method bears similarity to prior work but outperforms them on consistency score even with arguable disadvantage, it implies there is room for improvement on soundness for previous works.

Table 1 also shows that upscaling improves the consistency score. As mentioned in Section 4, upscaling was used in previous work to avoid "artifacts" and produce better looking masks. Our experiment shows a new explanation for the upscaling that it improves the soundness of the masking.

To verify the connection between completeness/soundness and consistency score, we compute the completeness and soundness (with $\epsilon = 10^{-3}$) for test samples, and report the average in Table 2. It shows completeness/soundness and consistency score are highly correlated. Even though saliency methods like Phang et al. (2020) have inherent advantage in soundness, our method still achieves the best soundness. Our method also gains better completeness, which is likely contributed by maximizing the probability assigned to the given label instead of matching the model probability.

Table 2: Completeness and soundness for each saliency methods, averaged over test samples. Completeness are computed on labels that the model predicts, and soundness are computed on the incorrect labels (w.r.t. model prediction). For soundness, we compute both the average among the incorrect labels and the minimum one among them. Consistency scores are listed in the last column for a comparison. The best two of each column are marked bold. Consistency scores are highly correlated to completeness/soundness in the table.

| | Completeness of model prediction | Average soundness of the incorrect labels | Worst soundness of the incorrect labels | Consistency Score |
|---|---|---|---|---|
| Gradient $\odot$ Input | 0.54 | **0.98** | 0.83 | 0.841 |
| Real time saliency | 0.56 | **0.98** | 0.85 | 0.857 |
| Fong & Vedaldi (2017) | 0.53 | 0.95 | 0.61 | 0.845 |
| Phang et al. (2020) | 0.71 | **0.98** | **0.87** | **0.871** |
| Ours ($s = 1$) | **0.85** | 0.95 | 0.61 | 0.862 |
| Ours ($s = 4$) | **0.83** | **0.99** | **0.89** | **0.880** |
| Random | 0.35 | 0.96 | 0.70 | 0.829 |

Table 3: Consistency scores for different $\lambda_{TV}$ in CIFAR-10. The last two rows are the mask-model consistency conditioned on whether the base classifier predicts the ground truth. The best one in each row is marked bold. $\lambda_{TV} = 0.1$ performs the best in both situations.

| $\lambda_{TV}$ | 0 | 0.001 | 0.01 | 0.1 |
|---|---|---|---|---|
| Consistency Score | 0.46 | 0.37 | 0.49 | **0.62** |
| Consistency (correct) | 0.47 | 0.38 | 0.51 | **0.64** |
| Consistency (incorrect) | 0.24 | 0.21 | 0.16 | **0.36** |

## 6.2 EFFECT OF TV REGULARIZATION

In this subsection, we test our theoretical prediction from Section 5 that TV penalty improves soundness. We first visually inspect the masks learned by our procedure as we increase the TV regularization strength[8] in Figure 4. Masked images at 10 % sparsity are depicted in odd columns and the full masks are shown in even columns. We plot the AUC insertion metric with blur evaluated on the non-sparsely masked images on the $y$ axis of each plot. We find that as we increase the TV regularization strength, the model can still find a saliency map for the correct label with high AUC score, however the AUC score for the mask learned to fit the second most confident label drops significantly. This, in conjunction with Lemma 3.5, suggests that the TV regularization method has slightly worse *completeness* but much higher *soundness*. Figure 3 plots the model output probability for CIFAR-10 as more pixels from the original image are retained using the mask. Our high level finding is again that adding TV penalty and upsampling significantly aid soundness, while only slightly hurting completeness. To formally justify the effect of TV penalty, we measure our consistency score of different level of TV penalty on CIFAR-10. To further challenge the completeness and soundness, the salient sets of our consistency score follows the Insertion game but only takes sets of size $0.2d$ to $0.6d$ ($d$ is the number of pixels). Table 3 shows the result.

## 7 CONCLUSIONS

Saliency explanations of ML models has proved nebulous and generated many controversies. By taking rooted in intrinsic definitions such as completeness/soundness and consistency score, this paper has tried to provide greater rigor to the intrinsic approaches to saliency. Other new contributions include clarifying the role of TV regularization (it hurts completeness slightly but greatly improves soundness); extensive experimental evaluations that bring new understanding using consistency score; and a simple saliency method for producing mask-based explanations whose performance is broadly competitive with good existing methods, and sometimes better.

The soundness notion is clearly useful for *localization* of objects in the image, and for classification in the wild –where multiple objects appear over the image. This was hinted in our evaluation on a prior small dataset (see Appendix B). It may have other applications to handling distribution shift and other types of robustness, but such explorations are left for future work.

---

[8]Similar images for scaling factors can be found in Appendix C

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

# A    INTUITIONS AND IMPLEMENTATIONS OF PROCEDURES TO FIND MASKING EXPLANATIONS

As introduced in Section 3.3, for evaluation we may interest in random binary masks due to its connection to AUC, but in our method for finding masking explanations we only focus on deterministic masks. Given a network $f$, image $x \in \mathbb{R}^{c \times hw}$ and class $a$, we wish to find a binary mask $M \in \{0, 1\}^{hw}$ such that when the part of $x$ on $M$ is superimposed onto a distractor $\bar{x} \sim \mathcal{X}$ as $\tilde{x} = M \odot x + (1 - M) \odot \bar{x}$, the output probability of the model $f(\tilde{x}, a)$ is high for the class $a$. This corresponds to the case where $\Delta$ is a singleton that assigns probability 1 to $M$ and $\Gamma(x, M)$ is the distribution of $\tilde{x}^9$. As in Section 3.1 we compute the average probability assigned to class $a$ over the sampling of the distractor $\bar{x}$, i.e. we are interested in making $\mathbb{E}_{\bar{x} \sim \mathcal{X}}[f(\tilde{x}, a)]$ high. To avoid the hard problem of optimizing over the hypercube $\{0, 1\}^{hw}$, a typical strategy (also employed in prior work) is to relax the domain of $M$ to be $[0, 1]^{hw}$. Since we do not wish to learn masks of very large size, a $\ell_1$ norm penalty on $M$ (corresponding to size of the mask), leading to the following natural objective function[10]

$$L(M) = \mathbb{E}_{\bar{x} \sim \mathcal{X}} \left[ - \log(f(M \odot x + (1 - M) \odot \bar{x}, a)) \right] + \lambda_1 \|M\|_1 \tag{5}$$

However most masking-based methods employ additional "tricks" in order to avoid "artifacts" in the produced saliency maps, like Total-Variation (TV) penalty (Fong & Vedaldi, 2017) and upscaling of the mask from a lower resolution one (Petsiuk et al., 2018). We also employ the same strategy by learning a low-resolution mask at scale $s$, $M \in \mathbb{R}^{hw/s^2}$, to minimize the following

$$L(M) = \mathbb{E}_{\bar{x} \sim \mathcal{X}} \left[ - \log(f(M^{\times s} \odot x + (1 - M^{\times s}) \odot \bar{x}, a)) \right] + \lambda_{TV} TV(M^{\times s}) + \lambda_1 \|M^{\times s}\|_1 \tag{6}$$

where $M^{\times s} \in \mathbb{R}^{hw}$ is obtained by upscaling $M$ by a factor of $s \in \{1, 4\}$ via bilinear interpolation.

While the motivation cited for these "trick" is to avoid artifacts, it is not clear whether artifacts are a bad thing, since they might be relevant to the net's decision-making. Indeed, we show that while TV penalty or upscaling does produce better looking masks, they lead to a drop in the *completeness metric*. However we show that adding such tricks leads to significant improvement in the *soundness metric*, thus providing a novel justification for the use of such tricks, beyond just the heuristic argument of getting rid of artifacts. In Section 5 we also provide theoretical justification for why TV penalty can help with soundness, even for the simple case of linear predictors on non-image data.

We optimize the objective in Equation (4) by parametrizing $M$ as a sigmoid of real valued weights $W \in \mathbb{R}^{hw/s^2}$, i.e. $M = \sigma(W)$, and run Adam (Kingma & Ba, 2014) optimizer for 2000 steps with learning rate 0.05 and by sampling 10 distractor images at every step, for different values of $\lambda_{TV}$ and upscaling factor $s$. We report the effect of $\lambda_{TV}$ qualitatively in Figure 4 and quantitatively on various intrinsic saliency metrics in Table 10.

# B    PRACTICAL BENEFITS OF SOUNDNESS FOR IMAGES OF MULTIPLE OBJECTS

Images may have multiple plausible labels. In Figure 2, images that previously used in Gu et al. 2018 can have both elephants and zebras present, but it may not be always clear from the model output if there is such a case, since the model can be much more confident on one label, e.g., elephant, than one would expect it to be. For this reason, finding masking explanations validating other labels, e.g., zebra, could provide more information on how the model makes the prediction.

# C    EXPERIMENTAL DETAILS AND ADDITIONAL EXPERIMENTS

In this section we expand upon the experiments in Section 6 and complement them with more experiments on the ImageNet, CIFAR-10 and CIFAR-100 datasets. For each of the datasets we test the following:

---

[9]We slightly abuse the use of notation to use $M$ as a set over coordinates rather than a binary mask.

[10]A standard way to maximize probability is to minimize the negative log probability

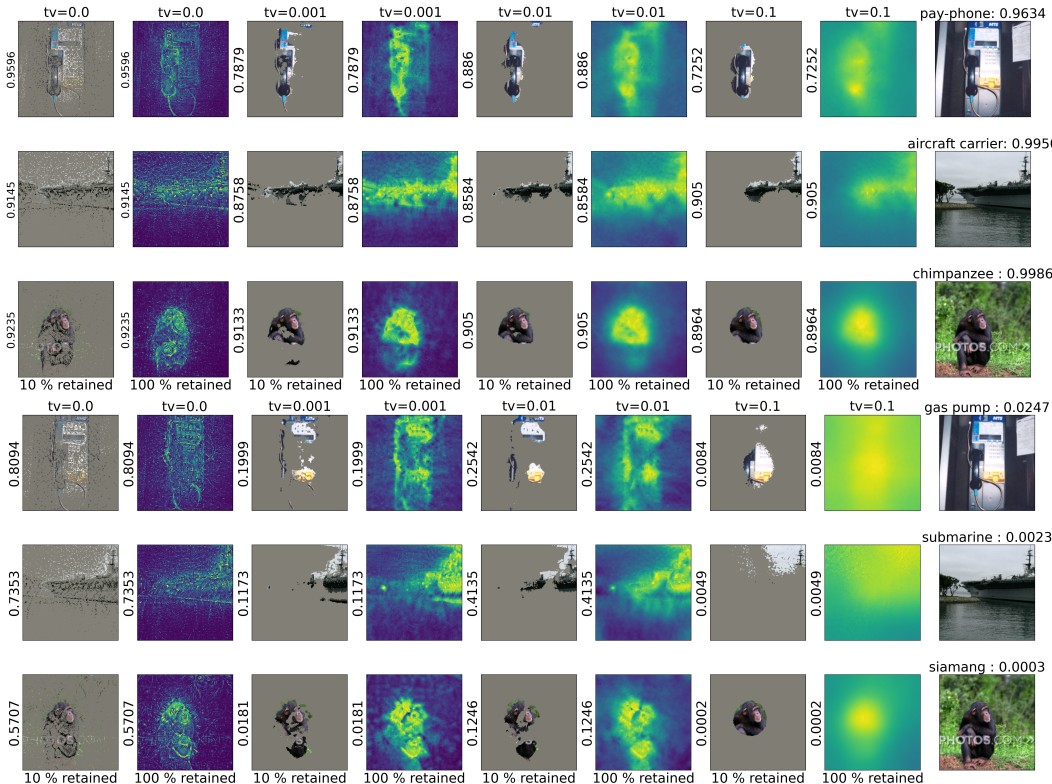

Figure 4: **Top**: Masking validating the correct label of ImageNet images using Section 4. Pixels outside the salient set $S$ are rendered as grey. Size of $S$ (as % of total pixels) appears below the images. Value of TV regularizer shown above each column corresponding to (0,.001,.01,.1). Original image rendered in last column. True labels with original model probabilities are shown in the rightmost column. Insertion metric when the mask is found by the procedure in Section 4 shown on y axis **Bottom**: Masking validating the second-best label, which appears on the right along with original model probability of that label. Insertion metric when the mask is found by fitting the second best label under the procedure in Section 4 is shown on the y label of each plot. We find that increasing the TV regularizer makes the resulting assignment more sound.

- **Visualization:** For various values of TV regularization (and upscaling for ImageNet), we visualize the mask and also what part of the image a sparse version of the mask highlights. We do so for masks learned for the correct label and also for the second most probable label as predicted by the model. The common trend is that while TV regularization (and upscaling) make the masks more human interpretable, it also makes it harder to find a good mask (partial assignment) for the incorrect label, thus improving soundness.

- **AUC curve:** We plot the output model probability for a masked input as more pixels from the original image are selected. The 4 plots denote replacing remaining pixels with grey pixels or pixels from a random image, and masks to fit the correct or incorrect label. Again, we find the TV regularization and upsampling help with soundness. For mask $M$, if $\bar{M}(p)$ denotes the discrete mask with top $p$ fraction of the pixels from $M$ picked. We plot $\mathbb{E}_x\left[\mathbb{E}_{x'\sim\Gamma}[f(\bar{M}(p)\odot x + (1-\bar{M}(p))\odot x', a)]\right]$ v/s $p$, where $\Gamma$ is either a random image or a grey image, and $a$ is either the correct label for $x$ or the second best label.

- **Completeness/soundness:** We evaluate completeness and soundness metrics as defined in Definition 3.2. In particular, for every input $x$, we evaluate completeness for the correct label $a$ and soundness for the incorrect (second best) label $a'$. For any mask $M$, we define AUC as $AUC(M;(x,a)) = \mathbb{E}_p\left[\mathbb{E}_{x'\sim\Gamma}[f(\bar{M}(p)\odot x + (1-\bar{M}(p))\odot x', a)]\right]$. Com-

pleteness and soundness are defined as

$$C_\delta(M) = \mathbb{E}_{(x,a)} \left[ \min \left\{ \frac{AUC(M;(x,a))}{\min\{f(x,a),\delta\}}, 1 \right\} \right] \tag{7}$$

$$S_\epsilon(M) = \mathbb{E}_{(x,a')} \left[ \min \left\{ \frac{\max\{f(x,a'),\epsilon\}}{AUC(M;(x,a'))}, 1 \right\} \right] \tag{8}$$

where $a$ is the correct label and $a'$ is the incorrect label and $f(x,a)$ is the model probability for label $a$ for input $x$. $\delta$ and $\epsilon$ can be any reasonable constant to stabilize the value.

- **Intrinsic metrics:** We evaluate our masks on other intrinsic metrics from prior work, and compare to baseline saliency methods. Our baselines include Gradient $\odot$ Input (Shrikumar et al., 2017), Smooth-Grad (Smilkov et al., 2017), Real Time Saliency (Dabkowski & Gal, 2017) (for ResNet-50 on ImageNet), and Random indicating a random Gaussian mask as a control. We use Captum (Kokhlikyan et al., 2020) for Gradient $\odot$ Input and Smooth-Grad implementations and the original author code[11] for Real Time Saliency. When calculating the Saliency Metric (SM) (Dabkowski & Gal, 2017) we tune the threshold $\delta$ on a holdout set of size 100 with $\delta$ between 0 and 5 in increments of 0.2 as in prior work.

  For the saliency method of Fong & Vedaldi (2017) that we only used on the Imagenette, we adapt the most popular implementation on GitHub[12]. The implementation contains minor deviations from the original paper as described on its main page. For Phang et al. (2020), we used their best CA model pretrained and provided in original author code[13].

## C.1 CIFAR-10 Experiments

We also run our method from Section 4 on the CIFAR-10 dataset using a pretrained ResNet-164 architecture[14]. For all experiments we learn a mask $M \in \mathbb{R}^{32 \times 32}$, thus using a scaling factor of $s = 1$ (no upscaling). We train masks for 1600 images that were correctly classified by the pretrained ResNet-164 using regularization parameter $\lambda_{TV} \in \{0, 0.001, 0.01, 0.1\}$. We use a (fixed) L1 regularization value of .001 for all masks.

We visualize the masks learned for the correct label in Figure 6a and in Figure 6b we visualize the same for the second best label predicted by the ResNet-164 model. We also visualize the masks for all labels for some randomly picked images in Figure 7 to demonstrate the commonness of artifact, especially for the incorrect labels. The AUC curves in Figure 5 suggest a similar trend to that of ImageNet, adding TV regularization results in only a mild drop in completeness, but significantly improves soundness. Evaluation of our masks, compared to some gradient baselines, on intrinsic metrics can be found in Table 5. We place a downarrow after the name of the metric to indicate a lower value is considered better and an uparrow when a higher value is considered better. We evaluate on a randomly selected subset of 1000 data points where the model had correct top 1 accuracy. We report the completeness and soundness results for CIFAR-10 in Table 4 for TV values in (0.0,0.001,0.01,0.1) calculated using a ResNet-164 model.

## C.2 CIFAR-100 Experiments

We run the same experiment for CIFAR-100 using the corresponding ResNet164 model. We visualize the masks learned for the correct label in Figure 9a and in Figure 9b we visualize the same for the second best label predicted by the ResNet-164 model. The AUC curves in Figure 13 suggest a similar trend to that of ImageNet, adding TV regularization results in only a mild drop in completeness, but significantly improves soundness. Evaluation of our masks, compared to some gradient baselines, on intrinsic metrics can be found in Table 7. We place a downarrow after the name of the metric to indicate a lower value is considered better and an uparrow when a higher value is considered better. We evaluate on a randomly selected subset of 1000 data points where the model had correct top 1 accuracy. When calculating the saliency metric we tune the threshold $\delta$ on a holdout set of size 100 with $\delta$ between 0 and 5 in increments of 0.2.

---

[11]https://github.com/PiotrDabkowski/pytorch-saliency

[12]https://github.com/jacobgil/pytorch-explain-black-box

[13]https://github.com/zphang/saliency_investigation

[14]https://github.com/bearpaw/pytorch-classification. The ResNet-110 model in this repository is actually a ResNet-164 model.

Table 4: Completeness and soundness for a ResNet-164 CIFAR-10 as defined in Equation (8). Each column contains a tuple (Grey/Noise, TV 0.0/ TV 0.001/ TV 0.01/ TV 0.1). Grey indicates pixels were greyed during calculation. Noise indicates they were replaced with other images. TV indicates a TV regularization value of 0.0, 0.001, 0.01, or 0.1.

| Grey | TV = 0.0 | TV= 0.001 | TV= 0.01 | TV= 0.1 |
|---|---|---|---|---|
| Correct label completeness ($C_{0.8}$) | 0.92 | 0.91 | 0.84 | 0.79 |
| Second label soundness ($S_{0.2}$) | 0.27 | 0.28 | 0.34 | 0.38 |
| Noise | TV = 0.0 | TV= 0.001 | TV= 0.01 | TV= 0.1 |
| Correct label completeness ($C_{0.8}$) | 0.90 | 0.88 | 0.81 | 0.75 |
| Second label soundness ($S_{0.2}$) | 0.30 | 0.31 | 0.37 | 0.42 |

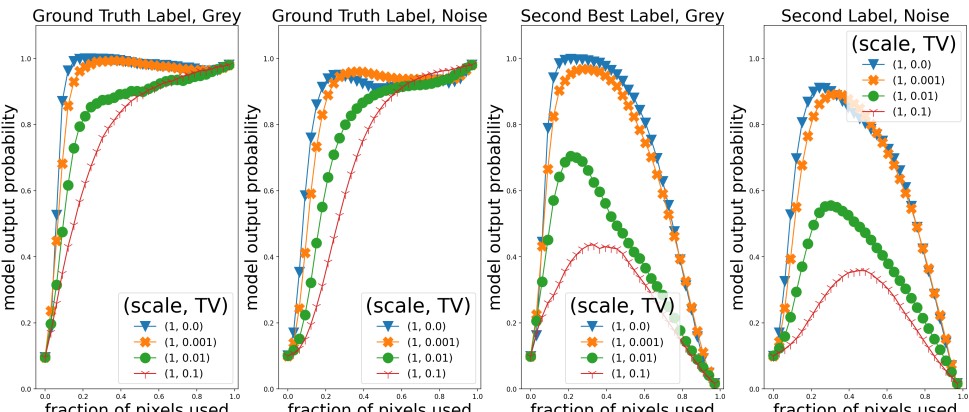

Figure 5: [CIFAR-10] AUC curves with as the fraction of pixels retained from the original images based on the mask varies from 0 to 1.0 on the X-axis. The probabilities assigned by the model (averaged over 1600 images) on the Y-axis. **Left:** Mask learned for ground truth label, probabilities for ground truth label while replacing remaining pixels with grey. **Center Left:** Mask learned for ground truth label, probabilities for ground truth label while replacing remaining pixels with other image pixels. **Center Right:** Mask learned for second best label, probabilities for second best label while replacing remaining pixels with grey. **Right:** Mask learned for second best label, probabilities for second best label while replacing remaining pixels with other image pixels. We see that increasing TV regularization results in only a mild drop in completeness, but significantly improves soundness.

Table 5: Performance of our method on CIFAR-10 and some baselines on various intrinsic saliency metrics proposed in prior work. We find that while both our masks (learned with and without TV) have very good performance on the insertion metric. The deletion and saliency metrics are uninformative in this case, since all methods are as good (or worse) compared to a random mask.

| | Gradient $\odot$ Input | Our method ($\lambda_{TV} = 0.01$) | Our Method ($\lambda_{TV} = 0$) | Smooth-Grad saliency | Random |
|---|---|---|---|---|---|
| Deletion $\downarrow$ | 0.32 | 0.37 | 0.59 | 0.31 | 0.26 |
| Insertion (blur) $\uparrow$ | 0.60 | 0.88 | 0.94 | 0.66 | 0.36 |
| Insertion (grey) $\uparrow$ | 0.51 | 0.83 | 0.92 | 0.55 | 0.26 |
| Saliency Metric $\downarrow$ | 0.22 | 0.22 | 0.22 | 0.23 | 0.22 |

We report the completeness and soundness results for CIFAR-100 in Table 6 for TV values in (0.0,0.001,0.01,0.1) calculated using a ResNet-164 model.

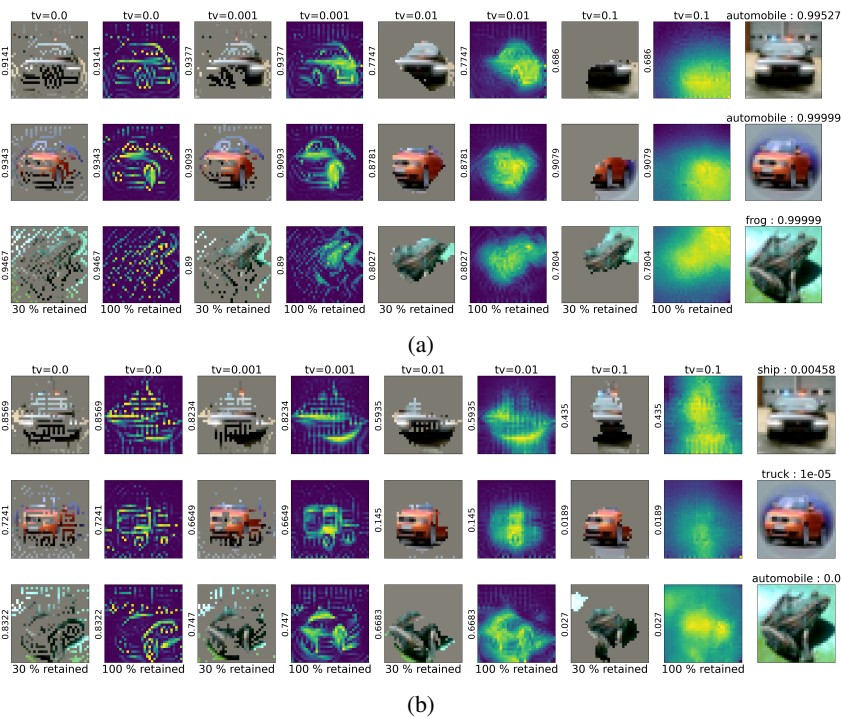

Figure 6: Details in Appendix C.1 **Panel 6a** Partial statistical assignments validating the correct label of CIFAR-10 images using the procedure outlined in Section 4 on ResNet-164. Columns (1,3,5,7) depict masked images at 30 (retained) % mask sparseness. Columns (2,4,6,8) depict the original mask. TV values shown above. Original image shown in rightmost column. Model probability of correct label for masked images on y axis. **Panel 6b** Partial statistical assignments validating the second most probable label of CIFAR-10 images using the procedure outlined in Section 4 on ResNet-164. Columns (1,3,5,7) depict masked images at 30 % mask sparseness. Columns (2,4,6,8) depict the original mask.TV values shown above. Original image shown in rightmost column. Model probability of second best label for masked images on y axis.

Table 6: Completeness and soundness for a ResNet-164 CIFAR-100 as defined in Equation (8). Each column contains a tuple (Grey/Noise, TV 0.0/ TV 0.001/ TV 0.01/ TV 0.1). Grey indicates pixels were greyed during calculation. Noise indicates they were replaced with other images. TV indicates a TV regularization value of 0.0, 0.001, 0.01, or 0.1.

| Grey | TV = 0.0 | TV= 0.001 | TV= 0.01 | TV= 0.1 |
|---|---|---|---|---|
| Correct label completeness ($C_{0.8}$) | 0.78 | 0.74 | 0.66 | 0.60 |
| Second label soundness ($S_{0.2}$) | 0.39 | 0.42 | 0.50 | 0.54 |
| Noise | TV = 0.0 | TV= 0.001 | TV= 0.01 | TV= 0.1 |
| Correct label completeness ($C_{0.8}$) | 0.72 | 0.70 | 0.65 | 0.58 |
| Second label soundness ($S_{0.2}$) | 0.46 | 0.47 | 0.51 | 0.56 |

## C.3    EXPERIMENTS ON IMAGENET

### C.3.1    RESNET-18

In Figure 10a we depict the the masks for TV values in $\{0.0, 0.01\}$ for a ResNet-18 model on ImageNet for the ground truth label and in Figure 10b we depict the same for the second best label. We also experiment with the effect of upsampling (US) the mask, whereby we learn a mask of size (56,56) and upsample to size (224,224). We use a fixed L1 regularization value of 2e-5. We depict our results on ImageNet and ResNet-18 in Table 9

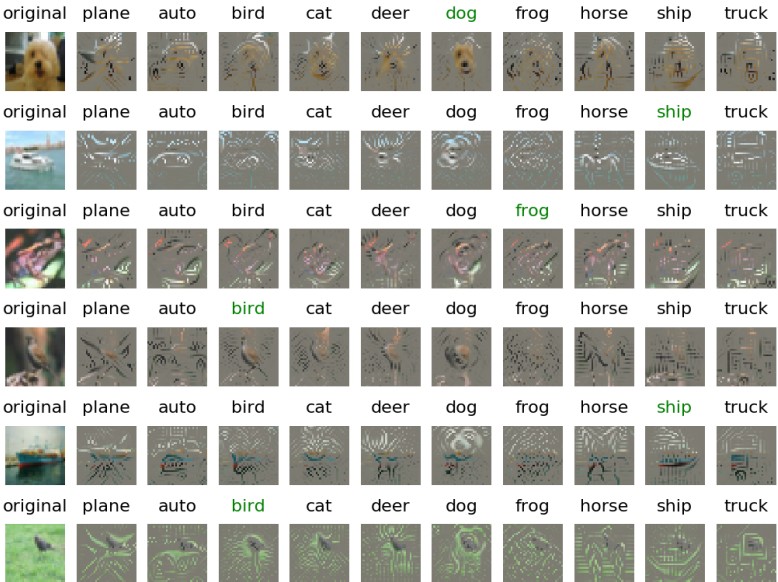

(a) Our method with no TV regularization

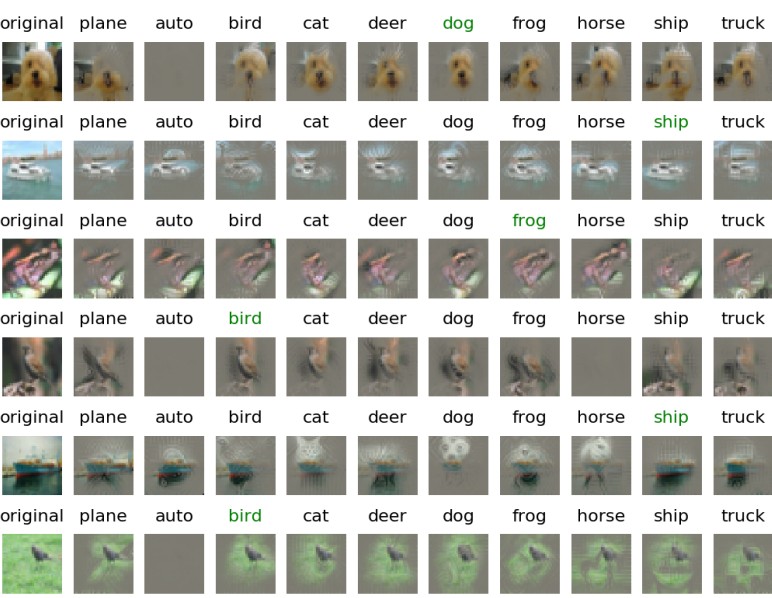

(b) Our method with TV regularization $\lambda_{TV} = 0.01$

Figure 7: A demonstration of artifacts created by masking. Pixels (partially) masked out are filled with gray based on the fractions they are masked out. Masks generated without or only with low level regularization can easily produce artifacts. It is more common and/or severe for the incorrect label than correct label.

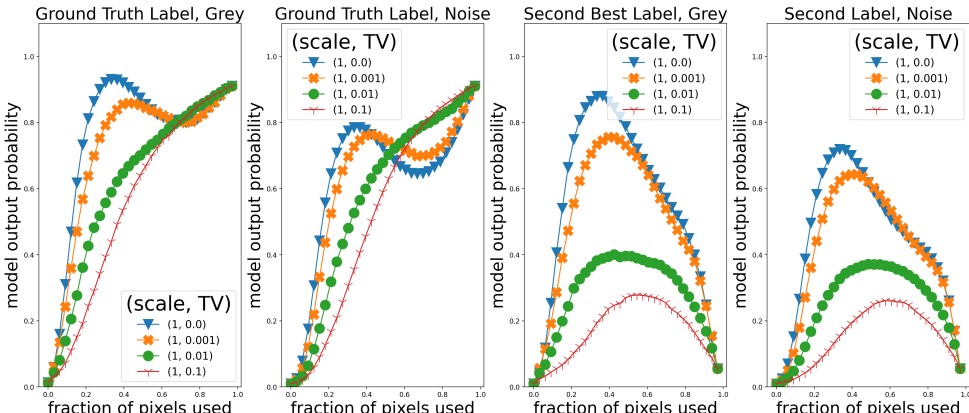

Figure 8: [CIFAR-100] AUC curves with as the fraction of pixels retained from the original images based on the mask varies from 0 to 1.0 on the X-axis. The probabilities assigned by the model (averaged over 1600 images) on the Y-axis. **Left:** Mask learned for ground truth label, probabilities for ground truth label while replacing remaining pixels with grey. **Center Left:** Mask learned for ground truth label, probabilities for ground truth label while replacing remaining pixels with other image pixels. **Center Right:** Mask learned for second best label, probabilities for second best label while replacing remaining pixels with grey. **Right:** Mask learned for second best label, probabilities for second best label while replacing remaining pixels with other image pixels. We see that increasing TV regularization results in only a mild drop in completeness, but significantly improves soundness.

Table 7: Performance of our method on CIFAR-100 and some baselines on various intrinsic saliency metrics proposed in prior work. We find that while both our masks (learned with and without TV) have very good performance on the insertion metric. The deletion and saliency metrics are uninformative in this case, since all methods are as good (or worse) compared to a random mask.

|  | Gradient $\odot$ Input | Our method ($\lambda_{TV} = 0.01$) | Our Method ($\lambda_{TV} = 0$) | Smooth-Grad saliency | Random |
|---|---|---|---|---|---|
| Deletion $\downarrow$ | 0.10 | 0.17 | 0.10 | 0.29 | 0.11 |
| Insertion (blur) $\uparrow$ | 0.36 | 0.71 | 0.82 | 0.39 | 0.20 |
| Insertion (grey) $\uparrow$ | 0.27 | 0.62 | 0.76 | 0.29 | 0.11 |
| Saliency Metric $\downarrow$ | 0.77 | 0.77 | 0.77 | 0.79 | 0.77 |

For the deletion metric, we note that most methods have comparable or worse performance than the random mask, which suggests that the metric does not give us much signal about the goodness of the saliency maps. On the insertion metric, we find that mask learned by not adding the TV penalty significantly beats other methods. The mask learned using TV penalty, on the other hand, has impressive performance on both the insertion AUC and saliency metric (SM).

**Completeness and Soundness on ImageNet and ResNet-18** We report our results in Table 8 for TV values in (0, 0.01) for both greying (Grey) and replacing with other image pixels (Noise). Additionally, we investigate the effect of upsampling (US) where we derive a (56,56) and upsample by a factor of 4 to a $(224, 224)$ mask.

**Effect of ensembling.** In order to investigate the effect of ensembling we plot maps in Figure 12 as we vary the number of maps that are ensembled over as $K \in \{1, 2, 4\}$, where we learn multiple masks such that each of them are individually statistical assignments. We do not upsample (using a scale of 1.0) and we use a fixed L1 regularization of 2e-5 and a fixed TV regularization of 0.0.

### C.3.2    RESNET-50

We present our results on ImageNet and ResNet-50 in Table 10. Using the same pretrained ResNet-50 model as Dabkowski & Gal (2017) lets us compare our method to their real-time saliency method

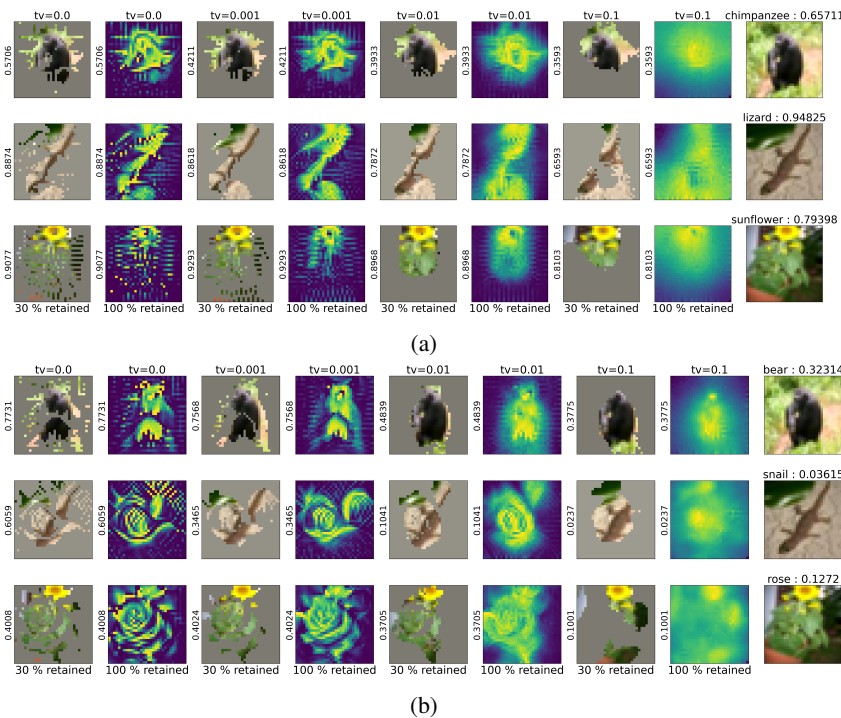

Figure 9: Details in Appendix C.2 **Panel 9a** Partial statistical assignments validating the correct label of CIFAR-100 images using the procedure outlined in Section 4 on ResNet-164. Columns (1,3,5,7) depict masked images at 30 (retained) % mask sparseness. Columns (2,4,6,8) depict the original mask. TV values shown above. Original image shown in rightmost column. Model probability of correct label for masked images on y axis. **Panel 9b** Partial statistical assignments validating the second most probable label of CIFAR-100 images using the procedure outlined in Section 4 on ResNet-164. Columns (1,3,5,7) depict masked images at 30 % mask sparseness. Columns (2,4,6,8) depict the original mask.TV values shown above. Original image shown in rightmost column. Model probability of second best label for masked images on y axis.

Table 8: Completeness and soundness for a ResNet-18 model on ImageNet as defined in Equation (8). Each column contains a tuple (Grey/Noise, TV 0.0/ TV 0.001/ TV 0.01/ TV 0.1). Grey indicates pixels were greyed during calculation. Noise indicates they were replaced with other images. no US indicates the full (224,224) mask was derived. US indicates a $(56, 56)$ mask was derived then upsampled by a factor of $4$. TV indicates a TV regularization value of 0.0 or 0.01.

| Grey | TV = 0.0 | TV = 0.01 | US TV = 0.0 | US TV = 0.01 |
|---|---|---|---|---|
| Correct label completeness $(C_1)$ | 0.93 | 0.82 | 0.75 | 0.72 |
| Second label soundness $(S_0)$ | 0.21 | 0.39 | 0.51 | 0.57 |
| Noise | TV = 0.0 | TV = 0.01 | US TV = 0.0 | US TV = 0.01 |
| Correct label completeness $(C_1)$ | 0.85 | 0.71 | 0.61 | 0.60 |
| Second label soundness $(S_0)$ | 0.24 | 0.40 | 0.53 | 0.57 |

directly on intrinsic metrics. Dabkowski & Gal (2017) do not evaluate on insertion/deletion metrics, so we evaluate these using their pretrained mask model.

## C.4 EFFECT ON SANITY CHECKS

Inspired by (Adebayo et al., 2018) we randomize the last layer of a ResNet-18 network and visually inspect the resulting saliency maps in Figure 11. We find that the maps appear less coherent than those of a pre-trained model. We use a fixed L1 regularization of 2e-5 and depict maps with and without upsampling (US) at TV values of $(0, 0.01)$.

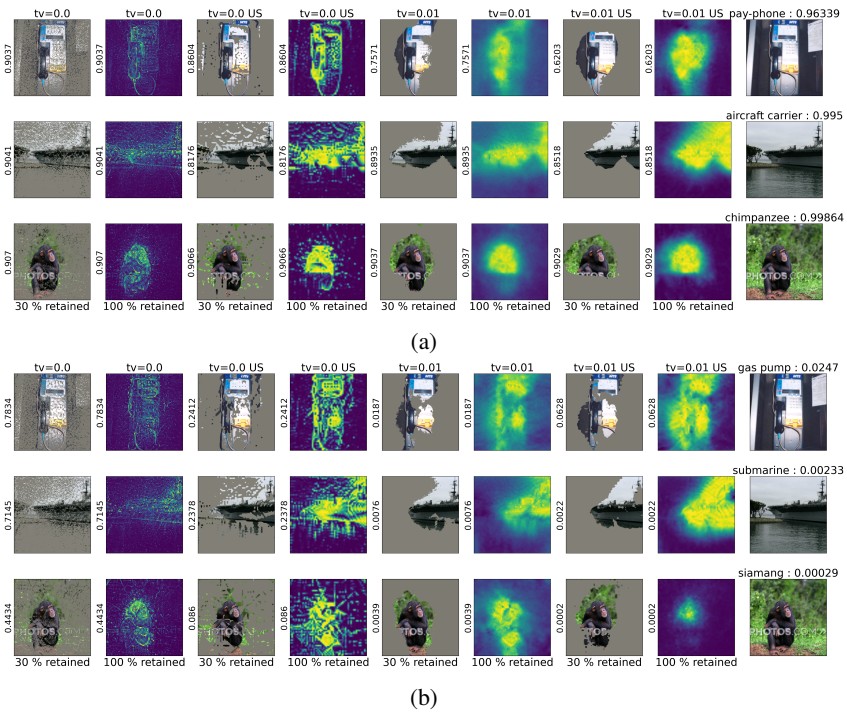

Figure 10: Details in Appendix C.3.1. US stands for upsampled mask, where we derive a (56,56) mask and interpolate to (224,224). **Panel 10a** Partial statistical assignments validating the correct label of ImageNet images using the procedure outlined in Section 4 on ResNet-50. Columns (1,3,5,7) depict masked images at 30 (retained) % mask sparseness. Columns (2,4,6,8) depict the original mask. TV values shown above. Original image shown in rightmost column. Model probability of correct label for masked images on y axis. **Panel 10b** Partial statistical assignments validating the second most probable label of ImageNet images using the procedure outlined in Section 4 on ResNet-50. Columns (1,3,5,7) depict masked images at 30 % mask sparseness. Columns (2,4,6,8) depict the original mask. TV values shown above. Original image shown in rightmost column. Model probability of second best label for masked images on y axis. We find, unsurprisingly, that adding TV regularization and upsampling make the mask more continuous and "human interpretable" and, more importantly, make it harder to find masks that can get high probability for the second best label, thus ensuring higher soundness.

Table 9: Performance of our method on ImageNet and ResNet-18 model and some baselines on various intrinsic saliency metrics proposed in prior work. We find that while both our masks (learned with and without TV) have very good performance on the insertion metric, the mask learned with TV has much better performance on the saliency metric. The deletion metric is uninformative in most cases, since most methods are as good (or worse) compared to a random mask.

| | Gradient $\odot$ Input | Our method ($\lambda_{TV} = 0.01$) | Our Method ($\lambda_{TV} = 0$) | Smooth-Grad saliency | Random |
|---|---|---|---|---|---|
| Deletion $\downarrow$ | 0.1054 | 0.1337 | 0.2080 | 0.0757 | 0.1336 |
| Insertion (blur) $\uparrow$ | 0.4443 | 0.7936 | 0.8507 | 0.5062 | 0.3118 |
| Insertion (grey) $\uparrow$ | 0.3056 | 0.6742 | 0.9213 | 0.3518 | 0.1325 |
| Saliency Metric $\downarrow$ | 0.3149 | 0.1507 | 0.3156 | 0.3157 | 0.3156 |

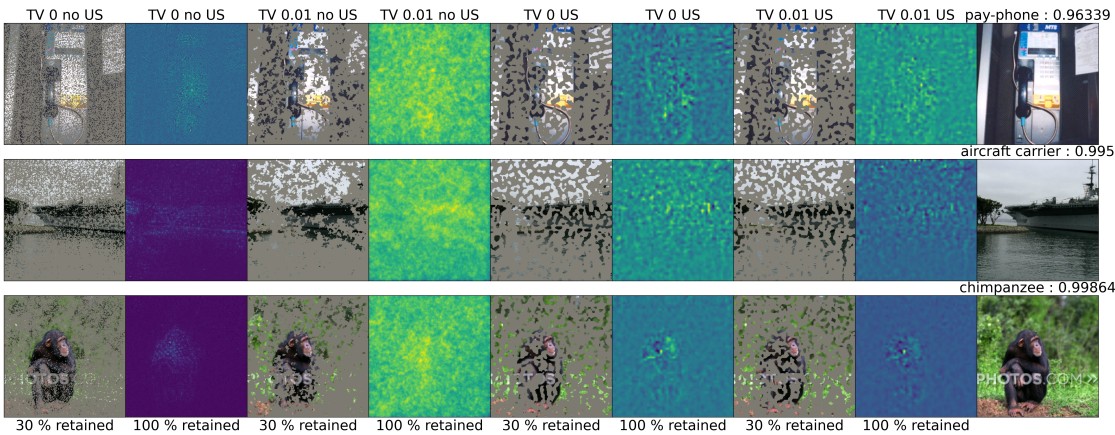

Figure 11: Results of randomizing the last layer of a ResNet-18 model on ImageNet data for the procedure described in Section 4. US indicates a $(56, 56)$ map was learned and upsampled to $(224, 224)$. We find the maps of this randomized network are less visually coherent than the analogous maps of a pre-trained model.

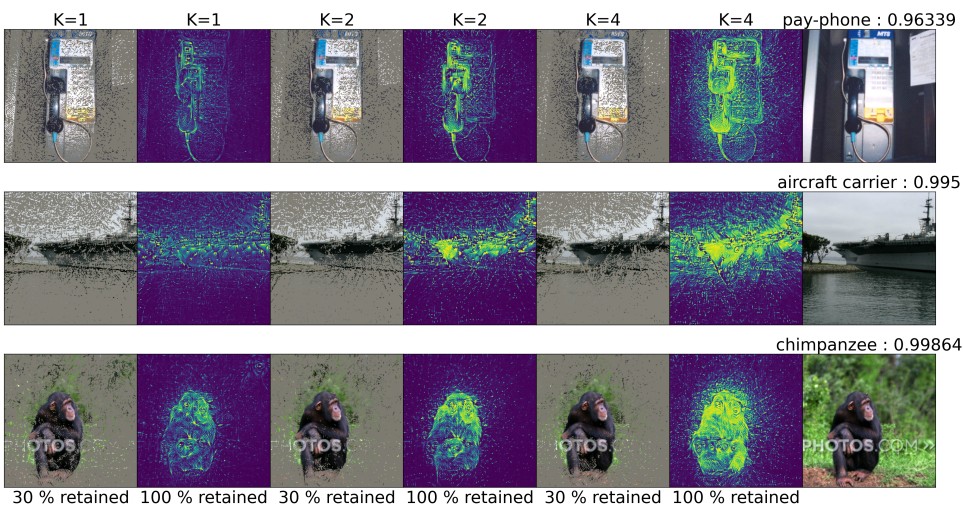

Figure 12: **Effect of ensembling** Partial statistical assignments validating the correct label of ImageNet and ResNet-18 images as we vary K, the number of maps. Details in Appendix C.3.1.

Table 10: Performance of our method on ImageNet and ResNet-50 model and some baselines on various intrinsic saliency metrics proposed in prior work. We find that while both our masks (learned with and without TV) have very good performance on the insertion metric, the mask learned with TV has much better performance on the saliency metric. The deletion metric is uninformative in most cases, since most methods are as good (or worse) compared to a random mask.

|  | Gradient $\odot$ Input | Our method $(\lambda_{TV} = 0.01)$ | Our Method $(\lambda_{TV} = 0)$ | Real Time saliency | Random |
|---|---|---|---|---|---|
| Deletion $\downarrow$ | 0.1451 | 0.1832 | 0.2067 | 0.1851 | 0.1843 |
| Insertion (blur) $\uparrow$ | 0.5434 | 0.8363 | 0.8673 | 0.6857 | 0.3562 |
| Insertion (grey) $\uparrow$ | 0.3716 | 0.7401 | 0.8857 | 0.4873 | 0.1835 |
| Saliency Metric $\downarrow$ | 0.2549 | 0.2019 | 0.2604 | 0.2943 | 0.2584 |

# D   ADDITIONAL BACKGROUND INFORMATION

## D.1   SALIENCY METHODS

We give a partial list of extant saliency methods here. We broadly categorize explanations into three categories: Back-propagation based explanations, axiomatic methods, and masking methods.

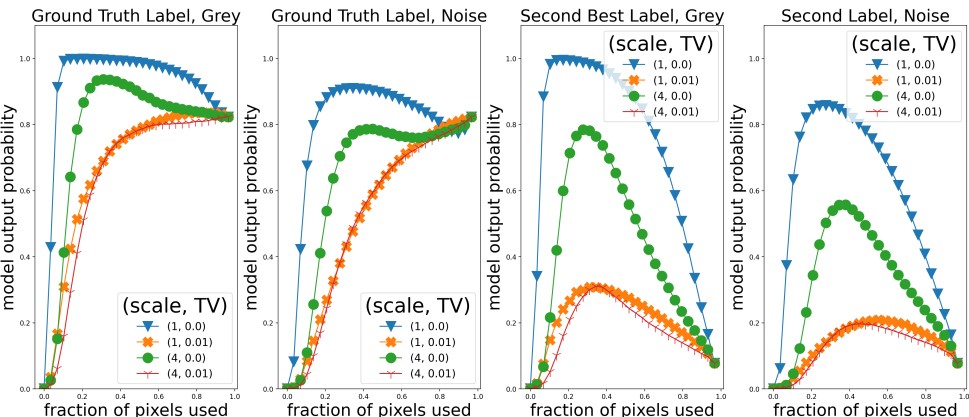

Figure 13: ImageNet data. **Left:** AUC curves as fraction of pixels used varies from 0 to 1.0 for ground truth label when replacing with grey. **Center Left:** AUC curves as fraction of pixels used varies from 0 to 1.0 for ground truth label when replacing with other images. **Center Right:** AUC curves as fraction of pixels used varies from 0 to 1.0 for second best label when replacing with grey. **Right:** AUC curves as fraction of pixels used varies from 0 to 1.0 for second best label when replacing with other images.

**Backpropagation based explanations** shape credit as it is propagated backwards through the neural network according to certain rules. These approaches include **Layerwise Relevance Propagation** (Binder et al., 2016) which satisfies completeness, **Rect-Grad** which thresholds internal neuron activations (Kim et al., 2019), and **DeepLIFT** which satisfies the summation to delta rule.

**Axiomatic methods.** Axiomatic methods decompose the ouput (typically the logit) according to certain axioms like fairness in Shapley based methods **SHAP** (Lundberg & Lee, 2017a) and **concept-SHAP** (Yeh et al., 2020). We also include gradient based approaches like **Gradient** $\frac{\partial S}{\partial x}$ (Baehrens et al., 2010) which calculates the partial derivative of the logit with respect to the input. **Gradient ⊙ Input** (Shrikumar et al., 2017) $\frac{\partial S}{\partial x} \cdot x$, which elementwise multiplies the gradient explanation by the input, and **Grad-CAM** (Selvaraju et al., 2019) which takes the gradient of the logit with respect to the feature map of the last convolutional unit of a DNN. **Smooth-Grad** (Smilkov et al., 2017), which averages the **Gradient ⊙ Input** explanation over several noisy copies of the input $x + \eta$, where $\eta$ is some Gaussian. The previous methods are intrinsic in the sense that they aim to explain the model decision. The last category of saliency maps, namely masking methods, also aim to explain the model decision, but they frequently aim to do so in a way that is interpretable by a human. Contrastive methods, such as contrastive layerwise propagation Gu et al. (2018), also modify LRP by constructing class specific saliency maps, with the goal of object localization, i.e. in an image of an elephant and zebra, the saliency map for elephant should have high overlap with the elephant, and similarly for the corresponding map for zebra.

**Masking Methods.** Masking Methods are often evaluated using a pointing game or WSOL metric, which measures overlap with human labeled bounding boxes or explanations. These masking methods include techniques based on averaging over randomly sampled masks (Petsiuk et al., 2018), optimizing over meaningful mask perturbations (Fong & Vedaldi, 2017), and real time image saliency using a masking network (Dabkowski & Gal, 2017). Pixels that have been removed from the image by the mask may be replaced by greying out, by Gaussian blurring as in Fong & Vedaldi (2017), or with infillers such as CA-GAN (Yu et al., 2018) used in Chang et al. (2018), or DFNet (Hong et al., 2019). De Cao et al. (2020) find masks using differentiable masking. Taghanaki et al. (2019) introduce a method that results in more accurate localization of discriminatory regions via mutual information.

**Pruning and information theory** Khakzar et al. (2019) improve attribution via pruning. Schulz et al. (2019) improve attribution by adding noise to intermediate feature maps.

**Saliency and Boolean Logic.** Previous work has also drawn connections between saliency and notions in logic. Ignatiev et al. (2019b) relates saliency explanations and adversarial examples by a

generalized form of hitting set duality. Ignatiev et al. (2019a) develops a constraint-agnostic solution for computing explanations for any ML model. Macdonald et al. (2019) develop a rate distortion explanation for saliency maps and prove a hardness result. Mu & Andreas (2020) find a procedure for explaining neurons by identifying compositional logical concepts. Zhou et al. (2018) describe network dissection, which provides labels for the neurons of the hidden representations. We are unaware of frameworks like Section 3.

**Arguments about saliency.** For discussion including pro/cons of various methods some starting points are Seo et al. (2018) Fryer et al. (2021) Gu et al. (2018) Sundararajan & Najmi (2020).

**Phang et al. (2020)** We describe separately the masking procedure used by Phang et al. (2020). They begin by taking a pretrained model on ImageNet. The masker has access to the internal representations of the pre-trained model, and tries to maximize masked in accuracy and masked out entropy. They do not provide the ground truth label to the masker.

## D.2 SALIENCY EVALUATION METHODS

Saliency evaluation methods attempt to evaluate the quality of a saliency map. Many interpret the heatmap values as a priority order of saliency. Extrinsic evaluation metrics include the **WSOL** metric, which aim to measure overlap of the saliency map with a human annotated bounding box and the **Pointing Game** metric proposed by Zhang et al. (2018) in which a pixel count as a hit if it lies within a bounding box and a miss otherwise, and the metric is $\frac{\text{\# Hits}}{\text{\# Hits} + \text{\#Misses}}$. Other more intrinsic methods include early saliency evaluation techniques like **MorF** and **LerF** Samek et al. (2016), which involve removing pixels either in the order of highest importance or lowest importance and observing the area of the resulting curve. **Insertion and Deletion Games** of Petsiuk et al. (2018) uses this too. The deletion game measures the drop in class probability as important pixels are removed, while the insertion game measures the rise in class probability as important pixels are added. (Our AUC discussion in Section 3 relates to this.) **Remove and Retrain (ROAR)** is a saliency evaluation method proposed by Hooker et al. (2019). Input features are ranked and then removed according to a saliency map. A new model is trained on the modified training set, and a larger degradation in accuracy on the modified test set compared to the original model on the original test set is regarded as a better saliency method. (NB: retraining makes this a non-intrinsic method.) Previous work has also introduced datasets specifically designed to test saliency methods. **BAM** Yang & Kim (2019) creates saliency maps by pasting object pixels from MSCOCO Lin et al. (2014) into scene images from MiniPlaces Zhou et al. (2017). The **Saliency Metric** proposed by Dabkowski & Gal (2017) thresholds saliency values above some $\alpha$ chosen on a holdout set, finds the smallest bounding box containing these pixels, upsamples and measures the ratio of bounding box area to model accuracy on the cropped image, $s(a,p) = \log(\max(a, 0.05)) - \log(p)$ where a is the area of the bounding box and p is the class probability of the upsampled image.

**Controversies.** There is extensive discussion of validity of saliency evaluation methods; e.g., Brunke et al. (2020)Petsiuk et al. (2018).

## D.3 SALIENCY COMPUTATIONS AND UNDERLYING MEANINGS OF SALIENCY

For simplicity this discussion assumes the datapoints are images and the classifier is a deep net. The heatmap in the saliency method is trying to highlight the contribution of individual pixels to the final answer. This is analogous to how a human may highlight relevant portions of the image with a plan. (Classic saliency methods in vision are inspired by studies of human cognition.) Saliency methods operationalize this intuitive definition in different ways, and we try to roughly categorise these as follows.

**Variational interpretation.** These interpret saliency in terms of effect on final output due to change in a single pixel –captured either via partial derivative of output with respect to pixel value (i.e., effect of infinitesimal change), or via change of output when this pixel is set to 0 or to a random (or "gray") value. Examples include **Gradient**, **Gradient ⊙ Input** Shrikumar et al. (2017), **Occlusion**

**Credit attribution guided by gradient.** These use the gradient to guide the assignment of saliency values. The gradient is interpreted as propagating values from the output to the input layer, and the values are partitioned/recombined at internal nodes of the net following some conservation princi-

ples. A key goal is to ensure *completeness*, which means that the sum of the attributions equal the logit value. Examples include **LRP**, **DeepLIFT** Shrikumar et al. (2017), **Rect-Grad** Let $a_i^l$ be the activation of some node in layer $l$, and $R_i^{l+1}$ be the backpropagated gradient up to $a_i^l$. Rect-grad replaces the vanilla chain rule, $R^l = \mathbf{1}[a_i > 0]$ with the rule that $R_i^l = \mathbf{1}[R_i^{l+1} a_i > \tau]$ for some threshold $\tau$. Hence, during a backward pass preference is given to nodes with large margin.

**Ensembling on top of above two ideas.** Ensembling methods combine saliency estimates over multiple inputs an an attempt to reduce noise in the final map. Examples include **Smooth-Grad**, Occlusion based methods, etc. We also include **Shapley Values** in this list.

The Shapley value aims to fairly distribute credit among a coalition of $N$ players. In the context of image saliency, each coordinate of the image input may be seen as a player, and the Shapley value computes $\sum_{S \subseteq N \setminus \{i\}} \frac{|S|!(n-|S|-1)!}{n!}(v(S \cup \{i\}) - v(S))$. It can be interpreted as the marginal contribution of player i, over all possible orderings of the coalition. In this sense, it can be seen as an ensembling method, as it averages over all possible random permutations.

**Analysis of saliency methods.** Previous work has analyzed ensembling methods like Smooth-grad, and found that it does not smooth the gradient Seo et al. (2018). They conclude that Smooth-Grad does not make the gradient of the score function smooth. Rather Smooth-grad is approximately the sum of a standard saliency map and higher order terms and the standard deviation of the Gaussian noise. It has also been found that Shapley values, despite having a uniqueness result, can differ in the way they depend on the model, data, etc Sundararajan & Najmi (2020). Fryer et al. (2021) highlight several nuances that should be taken into account when considering Shapley values. They introduce Shapley values as averaging over submodels, and note that "the performance of a feature across all submodels may not be indicative of the particular performance of that feature in the set of optimal submodels.". They provide specific cases where satisfying the axioms of Shapley values works against the goal of feature selection.

# E   CLARIFYING BENEFIT OF TV REGULARIZATION

This section illustrates Definitions 3.1 and 3.2 using linear classifiers. It also shows how TV regularizers help ensure soundness even in this setting.

Let $\mathcal{S}$ be a dataset of labeled data $(\boldsymbol{x}, y)$ where the inputs are of unit norm and labels are binary, i.e., $\|\boldsymbol{x}\|_2 = 1$, $y \in \{\pm 1\}$. The model in question is a linear classifier $f(\boldsymbol{x}) := \mathrm{sgn}(\langle \boldsymbol{w}, \boldsymbol{x} \rangle)$ parameterized by the weight vector $\boldsymbol{w} \in \mathbb{S}^{d-1}$, and it achieves the perfect accuracy on the set $\mathcal{S}$ with a margin $\gamma := \min_{(\boldsymbol{x}, y) \in \mathcal{S}} y \langle \boldsymbol{w}, \boldsymbol{x} \rangle > 0$. We assume that the coordinates of $\boldsymbol{x}$ and $\boldsymbol{w}$ are uniformly bounded by $\frac{10}{\sqrt{d}}$, i.e., $\|\boldsymbol{x}\|_\infty \leq \frac{10}{\sqrt{d}}$, $\|\boldsymbol{w}\|_\infty \leq \frac{10}{\sqrt{d}}$ (10 can be changed to any other constant).

Let $\Gamma$ be the input modification process that sets all non-salient pixels to 0. We are interested in masking explanations with deterministic salient set, i.e., $\Delta$ assigns probability 1 to some salient set $S$. According to Section 3.1, a masking explanation validates label $a$ if $\mathbb{E}_{\tilde{x} \sim \Gamma(x, \Delta)}[\mathbb{1}_{[f(\tilde{\boldsymbol{x}}) = a]}]$ is high. A simple calculation shows that this expectation equals to $\mathbb{1}_{[a \sum_{i \in S} w_i x_i > 0]}$, and thus the goal is to find $S$ so that $a \sum_{i \in S} w_i x_i > 0$.

As we do not consider the full salient set informative, we are interested in masking explanations with size constraint $|S| = L$ for some $1 \leq L \leq d$. There is a simple saliency method that achieves this goal: Given an input $\boldsymbol{x}$ and a label $a \in \{\pm 1\}$, sort the coordinates according to $a w_i x_i$ and take the highest $L$ coordinates as the salient set $S$.

It is easy to see that this method always produces $S$ with $a \sum_{i \in S} w_i x_i > 0$. Letting $a = y$ proves the completeness. However, this method does not satisfy soundness: a salient set $S$ with $a \sum_{i \in S} w_i x_i > 0$ can also be found for $a \neq y$!

Now we see how the TV constraint helps to ensure soundness (with good probability). A vector can be seen as a 1D image, and the TV of a salient set $S$ can be defined by $\mathrm{TV}(S) := \sum_{i=1}^{d-1} \left| \mathbb{1}_{[i \in S]} - \mathbb{1}_{[i+1 \in S]} \right|$. For simplicity, we consider salient sets with TV at most 2. This means $S$ is just an interval. Given the size and TV constraints $|S| = L$, $\mathrm{TV}(S) \leq 2$, it is easy to come out with the following saliency method: search over all the intervals of length $L$ and if an interval

$S$ satisfies $a \sum_{i \in S} w_i x_i > 0$, return it as the salient set. Fortunately, this method does satisfy both completeness and soundness, which is justified by the following theorem.

**Theorem E.1.** *For any $(\boldsymbol{x}, y) \in \mathcal{S}$, if we shuffle the coordinates of $\boldsymbol{w}$ and those of $\boldsymbol{x}$ according to the same random permutation, then*

1. *For $L = \Omega(\frac{1}{\gamma^2} \log \frac{1}{\delta})$, with probability $1 - \delta$, there is an interval of length $L$ that validates $y$;*

2. *For $L = \Omega(\frac{1}{\gamma^2} \log \frac{d}{\delta})$, with probability $1 - \delta$, no interval of length $L$ can validate $-y$.*

*Proof.* Let $S$ be any fixed interval of length $L$, then the distribution of $\sum_{i \in S} w_i x_i$ is identical to the distribution of the sum of $L$ i.i.d. random variables drawn from $\{w_1 x_1, \ldots, w_d x_d\}$ without replacement. Note that $dy w_1 x_1, \ldots, dy w_d x_d$ are $d$ numbers with mean $\gamma$, and their absolute values are bounded by $10^2 = O(1)$. By Chernoff bound,

$$\Pr\left[ \frac{1}{L} \sum_{i \in S} dy w_i x_i - \gamma < -\epsilon \right] \leq e^{-O(\epsilon^2 |S|)}.$$

Set $\epsilon = \gamma$ ensures that $y \sum_{i \in S} w_i x_i > 0$ with probability $e^{-O(\gamma^2 |S|)}$. We can fix any interval $S$ to prove Item 1. Taking union bounds over all intervals of length $L$, the probability of existing an interval of length $L$ that validates $-y$ should be no greater than $\sum_{|S|=L} e^{-O(\epsilon^2 L)} \leq d^2 e^{-O(\gamma^2 L)}$, which proves Item 2. □

This shows that such masking explanations make sense to humans: if the model predicts $y$, then we can find an interval of length $\tilde{\Omega}(1/\gamma^2)$ so that computing the inner product only in that interval leads to the same prediction; otherwise if the model does not predict $y$, such interval cannot be found. Thus it is sufficient to convince humans that the model predicts $y$ by only revealing the existence of such interval and the coordinate values in it.

Although the example given in this section is simple, the conceptual message is clear: saliency methods may not guarantee soundness in itself, but adding regularity constraints such as TV can mitigate this soundness issue effectively.

