# OpenReview forum: "New Definitions and Evaluations for Saliency Methods: Staying Intrinsic and Sound"
_ICLR.cc/2022/Conference — ICLR 2022 Submitted_

### Official Review · Reviewer_nxmx · 2021-10-30

**Correctness:** 2
**Technical Novelty And Significance:** 2
**Empirical Novelty And Significance:** 2
**Recommendation:** 3
**Confidence:** 3

**Main Review:**

Strengths:

- The paper is well structured, and it is easy to read.

- The proposed saliency explanation approach is simple and general. It leads to extensive results on three datasets: CIFAR-10, CIFAR-100, ImageNet.

- The analysis of TV regularization is novel and interesting.


Weaknesses:

- While I generally agree with the idea of evaluating and improving the soundness of saliency methods. The premise that the top-1 prediction is considered to be the only correct label and the second prediction is considered to be incorrect, is problematic. Indeed, a well-trained model can achieve high classification accuracy on iconic images and the top-1 prediction usually matches the ground truth. However, in practice, the top-1 prediction is likely to be wrong （as in Table 1), or there are multiple labels that can be correct (as in Figure 4). In the case of Table 1, I am not sure whether the wrong model prediction or the ground truth is considered to be correct. In the case of Figure 4, it is not clearly stated which is the correct label, elephant or zebra or both, and what the completeness and soundness scores of the compared methods are. To avoid confusion, the definition of correctness needs to be clarified. Again, determining what labels are correct or incorrect is critical, which may require a thorough discussion.

- Given this unclear definition, determining what labels are correct or incorrect will be critical. The metrics defined in Eq. (7)-(8) may need to be revised to consider multiple correct/incorrect labels.

- The experiments show that TV penalty or upscaling generate more smooth saliency maps, which leads to decreased completeness and improved soundness. Can this observation be generalized to other smoothing techniques, such as a simple Gaussian filter? If the effect is mainly attributed to the smoothness of saliency maps, this approach looks a bit trivial. Other baseline methods, such as 1) a single Gaussian blog placed in the image center, 2) a randomly sampled saliency map from other images, or 3) a bottom-up saliency map independent of the labels, should be added, because the current pixel-wise randomization approach does not consider the distribution of image data.

-  It would be interesting to quantitatively evaluate the performance of saliency methods on the ImageNet localization task, and to perform a correlation analysis between the completeness/soundness scores and the localization accuracy.



**Summary Of The Paper:**

This paper introduces two intrinsic evaluation criteria for saliency-based model explanation: completeness and soundness. While completeness indicates a saliency method's ability to justify the correct label,  soundness requires the method to be unable to find a masked input that significantly increases the probability of an incorrect label. A consistency score is introduced to simultaneously evaluate both completeness and soundness. Further, taking into account the soundness factor, a saliency method is proposed to maximize the probability for a given label, rather than all labels. Experiments demonstrate that the heuristic TV regularization method can help with soundness, as suggested by high consistency scores.



**Summary Of The Review:**

While I appreciate the motivation and the overall idea of measuring soundness of saliency, the definition of correctness and soundness is a bit unclear, and the effectiveness of the proposed method is not sufficiently demonstrated by the experiments. Considering the strengths and weaknesses of this paper, I don't think it is ready for publication at this moment.

---

> ### Author Response · Authors · 2021-11-22
> **Response**
>
> Dear Reviewer nxmx,
>
> Thanks for your detailed review. We have revised our paper according to your comments. Please find our response to your questions below.
>
> - “In the case of Table 1, I am not sure whether the wrong model prediction or the ground truth is considered to be correct.” “To avoid confusion, the definition of correctness needs to be clarified”
>
> > We define “correct label” as the top model prediction and other labels to be “incorrect” as mentioned in the introduction; we have made the discussion clearer in the revision. The consistency score is defined as the fraction of samples whose saliency score based prediction is consistent with the model prediction. It does not depend on the ground truth label. “Consistency (correct)” in Table 1 refers to the consistency score if we only consider samples where the base prediction matches the ground truth; we had misused the notation of correctness there and we have changed the column title in the revision to avoid confusion. It helps show the difference between our methods and existing methods.
>
> - “ In the case of Figure 4, it is not clearly stated which is the correct label, elephant or zebra or both.” “Again, determining what labels are correct or incorrect is critical, which may require a thorough discussion.” “The metrics defined in Eq. (7)-(8) may need to be revised to consider multiple correct/incorrect labels”
>
> > For Figure 4, multiple objects in one image is not formally considered in the paper, as we assume single object classification is the prior knowledge. A quick extension to the multi-object case is to define correct labels as the labels whose model probability exceeds a certain threshold. Even without the extended definition of correct labels, our completeness/soundness notion in Definition 3.2 with suitable $\alpha, \beta$ already allows explanations to exist for both elephant and zebra as long as the model probability for elephant and zebra are high enough. Eq. (7)-(8) in the appendix is a simplified version of Definition 3.2 assuming the task is single object classification and only one label is correct. Furthermore, experiments in the appendix only look at images that have been correctly classified by the model, which has been the standard thing to do in prior work.
>
> - Can observation about TV be extended to other smoothing techniques, such as a simple Gaussian filter?
>
> > There might be nothing special about TV or upscaling, we only consider it because it is a common technique used in prior work, with non-intrinsic justifications. Through our soundness metric we show that use of TV regularization and upscaling have intrinsic justifications too. Other techniques, whether they are smoothing techniques or not, may also improve the soundness, which will provide intrinsic justification to them as well.
>
> - What about other baseline maps?
>
> >We tested the suggested baseline of a single Gaussian is placed in the image center for a better comparison. The completeness, average soundness, worst soundness, and consistency scores of the Gaussian baseline on imagenette are 0.63, 0.97, 0.74, 0.867 respectively. The soundness of the Gaussian baseline is close to the random baseline, but the completeness of the Gaussian baseline is much better, even better than most of the previous methods, which is very likely because of the prior of the dataset, i.e., objects are mostly centered in the images. As a result, the consistency score of the Gaussian baseline is very good. However, our method (with $s = 4$) is still the best.
>
> > A saliency map independent of labels is exactly what Phang et al. (2020) outputs. They are considered as the state-of-the-art intrinsic saliency methods. Note again, Phang et al. (2020) do not “try hard” to produce good saliency maps for incorrect labels, which gives them advantage in soundness. We will add the Gaussian baseline and more baselines in later versions after we finish running on all four datasets.
>
> - Evaluate saliency methods on ImageNet localization task. Measure correlation between completeness/soundness scores and localization accuracy
>
> > As mentioned in the intro, our main motivation is in an intrinsic evaluation of saliency methods, that only judge their capability of producing good explanations for the model prediction, and not the ground-truth label. The localization task is “extrinsic” in that it is trying to match the output of saliency method to an explanation that a human would call meaningful, but that might have nothing to do with why the model predicted what it did. For example, the model may predict “zebra” based on one of the zebras in the image, so it would be unreasonable to localize all the zebras.

---

> > ### Comment · Reviewer_nxmx · 2021-11-29
> > **Response to Updates**
> >
> > I would like to thank the authors for their thorough responses and updates. Some of my major concerns have been clarified.
> >
> > The authors' comments explained what correctness means in the context of this paper. The motivation and importance of measuring soundness have also been highlighted in the updates. After reading the responses it is more clear to me that an intrinsic evaluation of saliency methods needs to take both completeness and soundness into account.
> >
> > However, given the current experimental results, I still doubt the effectiveness of the proposed soundness measure. It appears to be very sensitive to the smoothness of the saliency map and spatial prior of the dataset. Therefore, I think this paper needs more experiments to demonstrate this contribution before it is ready for publication.

---

### Official Review · Reviewer_4yoz · 2021-11-01

**Correctness:** 3
**Technical Novelty And Significance:** 2
**Empirical Novelty And Significance:** 2
**Recommendation:** 5
**Confidence:** 4

**Main Review:**

Strengths：
1.	The method proposed in this paper does not need to introduce additional manual labeling or training costs, which is an interesting and worth exploring perspective.
2.	The paper provides a lot of intuitive visualization results, which is helpful to guide readers' understanding.

Weaknesses:
1.	We believe that the example in Figure 1 is not comprehensive and does not explain the difference between completeness and soundness well. We want to know what will happen if the saliency maps of the category labels do not overlap?
2.	Section.3 mentioned that this paper uses the images of the training set to fill the pixels outside the saliency map area S. Wouldn't other objects in the background area interfere with the result unpredictably? Perhaps more explanation should be given here.
3.	From the results in Table 1, compared with the method in Phang et al.(2020), the proposed method has no special advantages.
4.	The data in Table 2 confuses me. The completeness of different methods is very different, but the soundness is very close, especially the second column of data. How should we understand this result?


**Summary Of The Paper:**

This paper discusses the completeness of saliency maps and proposes the concept of soundness. Based on this, they propose consistency score to predict the possibility that the quality of the saliency map is consistent with the classification probability. Besides, the author uses these definitions to implement the intrinsic evaluation of masked-based saliency methods.

**Summary Of The Review:**

The concept of the proposed method is innovative, but many details of the method have not been explained clearly. In addition, more persuasiveness is needed in the experimental performance.

---

> ### Author Response · Authors · 2021-11-22
> **Response**
>
> Dear Reviewer 4yoz,
>
> Thanks for your detailed review. We have revised our paper according to your comments. Please find our response to your questions below.
>
> - Fig. 1 doesn’t explain completeness vs soundness. What if maps don’t overlap?
>
> > Fig. 1 illustrates the saliency method (with small regularization) is complete but not sound. It outputs a good mask for the correct label, but can also output masks for the incorrect labels (cat/frog). Note that by “correct” label we mean the top label as predicted by the model. Cat/frog are not present in the original images, but are created artificially in the masked images. We are not sure what you mean by maps not overlapping. If you are referring to the situation where multiple objects in one image, we provide some evidence that completeness and soundness may be helpful to localize the object (Figure 2 in the revised version), but our main focus of this paper is single object classification.
>
> - Wouldn’t replacing other pixels with other image pixels affect the result?
>
> > Replacing with training set image pixels will indeed make it harder to find a mask. Despite this, we can learn a mask that predicts the correct label with high confidence (72% on average for imagenette dataset). Note that a “random” other image is selected to fill the pixels, and not all of them might change the model prediction by a lot. We believe that this added hardness of the task helps increase the robustness of the output, thus achieving better performance on various metrics.
>
> - In Table 1, compared to Phang et al., the proposed method has no special advantages.
>
> > We address this in our general response to the experimental observations.
>
> - In Table 2, completeness of methods is very different, but soundness is close. What does this mean?
>
> > It is because completeness may be “harder” than soundness since the random mask (the last row) can only achieve low completeness of  0.35 but high average/worst soundness of 0.96/0.70. The overall high value of the second column of data shows it is hard to find masking explanations for every incorrect label, but the third column shows it is not hard to find a mask for some incorrect label. Note again that saliency methods like Phang et al. (2020) do not produce good saliency maps for the labels other than model prediction, so we have to pick a suboptimal mask for the other labels in soundness evaluation, which gives them advantage. However, our method still achieves the best soundness.

---

> > ### Comment · Reviewer_4yoz · 2021-11-30
> > **Comment**
> >
> > The authors made a comprehensive response to all my comments. However, some of these explanations are still not very convincing to me. Moreover, from the comments of other reviewers and their responses, the paper still has a long way to go before it can be accepted for publication. Thus I would keep my initial rating unchanged.

---

### Official Review · Reviewer_LG7F · 2021-11-02

**Correctness:** 3
**Technical Novelty And Significance:** 2
**Empirical Novelty And Significance:** 2
**Recommendation:** 5
**Confidence:** 2

**Main Review:**

**Strengths:**

- The introduction of soundness and consistency score for use in evaluating saliency maps is a good idea and is explored in several experiments.
- The introduction did a good job of setting up the distinction between saliency methods and saliency evaluation metrics and laying the groundwork for the paper
- The related work gives good background on related saliency methods and evaluation methods; however, the paper could include more context for the past work and its relationship to the current paper, especially in the "arguments about saliency" and "controversies" sections. It would be helpful to have a brief description of what the main points of controversy are and perhaps a note about how this work relates to those arguments.
- The paper proposes soundness as a quantitative way of discovering artifacts in saliency-based approaches and back this up empirically r.e. TV regularization.

**Weaknesses:**
- The impact of the work is unclear. The new saliency method is similar to prior work and has mixed results when evaluated against other baselines (Table 1). The introduction of new saliency metrics has the potential to be useful, but I do not understand what new information soundness or consistency score provide that are missing from other measures; furthermore, they are expensive to compute (Section 6).
- I found the descriptions and equations for completeness and soundness--Definition 3.2 on page 4--to be difficult to understand. I think it would be useful for the paper to include an intuitive description of the equations, how they interacted with each other, and whether alpha and beta should be minimized or maximized. For example, it would be nice to follow up the definition with an intuitive explanation of these metrics to help the reader understand what they are capturing. It may also be preferable to break up these two concepts into separate definitions.
- This is my current understanding of completeness and soundness: alpha provides a lower bound and 1/beta an upper bound on the ratio of the probability assigned to class a on the masked vs the unmasked input. It is desirable to have both alpha and beta close to 1 (however, additional constraints must be applied to avoid the trivial case where the whole image is considered salient). Alpha and beta close to 1 imply that for all classes, the model's scores on the masked input are similar to its scores on the unmasked input. Is my understanding correct? If not, could the authors please clarify? If so, it may help to include such a discussion in the paper.
- The paper should justify why completeness is desirable. There is some discussion of this in the introduction, which ties increased soundness to reduction of artifacts in the mask itself. However, there are situations where it would make sense for different masks to significantly change a classifier output. Consider the case where a cat/dog classifier is trying to classify an image with both a cat and a dog in it, then masking out the dog should increase the probability of cat. Similarly, a non-robust model might output very different results in response to different masks. As such, does completeness measure the goodness of the saliency method or the robustness of the model?
- For the experiments described in Section 6.1, are the authors trying to validate the usefulness of their metric or prove the benefit of their saliency method? The proposed saliency method does not outperform other baselines for most metrics.

**Additional small notes:**
- The approach was inspired by logical proof systems, which is an interesting source of inspiration! But the connection is not clear--perhaps adding a brief description of how soundness is derived from logical proof systems or some references on this point would be helpful.
- It would be helpful to readers if the contributions were stated more clearly. The introduction contains a bulleted list of "Other contributions"; I recommend changing this to a list of the top contributions.

**Summary Of The Paper:**

This paper addresses masking-based saliency methods for providing visual model explanations. It defines several new metrics for evaluating saliency methods: "completeness", "soundness", and "consistency score". It proposes a saliency method that is a variant of prior work. The paper shows through experiments that TV regularization improves soundness of saliency-based explanations.

**Summary Of The Review:**

The contributions of the paper (the proposed saliency method and the new saliency metrics) are modest and their significance is not clearly explained in the paper or supported by experiments. The paper should explain the concepts of completeness, soundness, and consistency and justify why they are desirable. The experiments should support the main contributions: they should show that the proposed metrics are useful and that the proposed saliency method outperforms baselines on these metrics.

---

> ### Author Response · Authors · 2021-11-22
> **Response**
>
> Dear Reviewer LG7F,
>
> Thanks for your detailed review. We have revised our paper according to your comments. Please find our response to your questions below.
>
> - Impact is unclear. The connection to logical proof systems is not clear.
>
> > Please see our general response about the intuition of soundness, which includes the connection to logical proof systems.
>
> - Descriptions/equations are difficult to understand. Spend more time explaining them. What is alpha/beta?
>
> > Your understanding of alpha/beta is correct. We added explanations of the definitions in the revision right after Definition 3.2.
>
> - Why is completeness desirable?
>
> > Completeness is what is usually (implicitly) done in prior work (e.g., Section 3 in Dabkowski & Gal (2017) and masked-in objectives in Phang et al., 2020)). Completeness roughly means the model is able to predict the given label with relatively high probability by only salient pixels, which we believe is a necessary condition but clearly not sufficient. We also believe that completeness and soundness together is not sufficient. Apart from good completeness and soundness, one may pose other constraints on saliency methods or optimize for other metrics.
>
> - Are experiments in 6.1 validating the metric or the method?
>
> > Please see our general response about the experimental observations.

---

> > ### Comment · Reviewer_LG7F · 2021-11-23
> > **Response to updated draft**
> >
> > Thank you for your thorough response to reviewer comments and updates to the paper. The updates to the manuscript significantly clarify parts of the paper and address some of my comments:
> >
> > - The authors' comments explain the connection between logical proof systems and soundness. Updates in the introduction of the paper draft help explain why soundness is important. I think that including the first 3 sentences in the author's comment marked "connection to logic" in the body of the paper would also be helpful, since the connection to logic helps explain why "one must verify that a different mask could not induce the net to output a label corresponding to other logits"
> > - The "main contributions" section in the introduction clarifies the contributions of the paper—it puts the emphasis on consistency score as the primary metric to be used going forward and contextualizes the performance of the new proposed saliency method in the experiments section.
> > - The related work contains more discussion.
> > - Page 5 contains an intuitive explanation of alpha-completeness and beta-soundness, which help make this section easier to understand. (Thanks to the authors for confirming my understanding of these concepts.)
> > - The updates to the first paragraph of section 6 help situate the experimental results and clarify the impact of the results.
> >
> > However, I still have some reservations about the paper:
> >
> > - The experiments do not clearly support the claims in the paper about the proposed saliency method. The revised draft states that 1) the proposed approach is competitive with other methods and 2) it does better on the consistency score metric compared to previous methods. However, the data itself is somewhat ambiguous. Looking at Table 1, it looks like the model (Ours, s=1) is competitive with prior approaches, while the model (Ours, s=4) narrowly wins against Phang et al. for best consistency score (and soundness improvements in Table 2). The version of the model that shows soundness improvements appears to perform worse on traditional saliency metrics. Thus, it is difficult to conclude that "consistency score has no major conflicts with previous metrics".
> > - The consistency score metric is named as one of the main contributions of the paper. However, this metric is "computationally infeasible" for datasets with large numbers of classes, which limits the usefulness of the metric.
> >
> > I've updated my score to reflect that the authors' comments and changes to the paper have made it significantly clearer. Soundness is better motivated and the takeaways from the experimental section are better explained. However, given the limitations of the experiments and the proposed metric, I don't think it is ready for publication.

---

> > > ### Author Response · Authors · 2021-11-25
> > > **Response to further comments**
> > >
> > > Thank you for acknowledging the revision and updating your score. We would like to address the two points of reservation that you raised.
> > >
> > > - Firstly we would like to point out that even disregarding our method and our score, there is no single method that uniformly dominates all other methods on all other metrics. It would be unfair to expect any new method to do so, and it wasn’t our intention with the proposed method either. The main point of our method is to incorporate soundness into the mask learning procedure in a way that not only is reasonably effective on other metrics, but also improves soundness (directly as in Table 2 and indirectly through consistency score). We will clarify this point in later versions.
> > >
> > >     Secondly, as we have reiterated in our response, some methods (including Phang et al.) have a big advantage in the consistency score because their code does not produce good masks for “incorrect labels” and thus we end up using suboptimal masks for those labels, leading to higher consistency scores. Despite this advantage, our method has a better consistency score, especially for inputs that are incorrectly classified by the base model (last column in Table 1).
> > >
> > > - You mentioned that your primary reservation now is the high computational cost. We’d like to clarify why this is not big negative:
> > >     - Our saliency method (i.e., mask computation) is light-weight, and comparable to existing methods. You are referring to high cost in the *evaluation* of soundness. But this high cost is incurred by researchers/designers of saliency methods; end-users don’t need to evaluate soundness!  If soundness is accepted as a valuable property then saliency researchers should indeed test for it. Furthermore, some prior evaluation methods (e.g., [1]) require training a sequence of neural networks on the data which is also inefficient.
> > >     - The cost of consistency evaluation scales linearly with the number of labels/classes.  If the number of labels is large, the cost can be ameliorated by evaluating soundness only using the top-k labels as predicted by the model, which is what we do in the Appendix for ImageNet/CIFAR-100 evaluations. We will clarify this point in the main paper.
> > >
> > >
> > > [1] S. Hooker, D. Erhan, P. Kindermans, B. Kim. A Benchmark for Interpretability Methods in Deep Neural Networks. NeurIPS 2019

---

> > > > ### Comment · Reviewer_LG7F · 2021-11-29
> > > > **Response**
> > > >
> > > > Thank you for your further comments. However, I still have reservations about the paper and will keep my rating.

---

### Official Review · Reviewer_Ge5Z · 2021-11-02

**Correctness:** 3
**Technical Novelty And Significance:** 2
**Empirical Novelty And Significance:** 2
**Recommendation:** 6
**Confidence:** 3

**Main Review:**

The paper brings an interesting concept, i.e., soundness, to the model explainability research. There, however, remains a big motivation gap. Why do we need soundness? Given that we are interested to understand what causes the prediction output of a black-box model, why the source of wrong predictions shall matter?

The paper proposes a new score to evaluate the quality of heatmaps. The score is well designed and suited for the proposed approach. It, then, compares some methods with the proposed approach using the same evaluation score tailored for it's proposed algorithm. Thus, one could wonder, why are we lacking a careful comparison and discussions with other scores? Is it fair to compare multiple methods on a score tailored for one?

On the practicality of the method, based on the note in the paper about back-up plan of running other methods, is there any statistics of how often this could happen in the experiments? And how much this could be a risk to the applicability of the approach?

Overall, the paper is interesting. One caveat is that it refers to the supplement so often. It would be nice to read a paper that is self-contained.

**Summary Of The Paper:**

The paper tackles the problem of model explainibility from the perspective of heatmap-based visualization approaches. It argues that achieving a good heatmap to explain the model decision is possible by considering two notions, completeness and soundness, borrowed from logic. In fact, the completeness condition is what already considered during the formulation of similar methods. Thus the novelty lies on studying and analyzing the usefulness of soundness. The paper, thus, proposes a metric to evaluate both completeness and soundness that is used to learn a mask that explains the model prediction.

**Summary Of The Review:**

The paper is providing good insight about model explainability and proposes interesting concepts. Reading the paper, it seems the presentation, storyline and experiments could be improved. I find it a bit too much scattered between the supplement and main paper.

---

> ### Author Response · Authors · 2021-11-22
> **Response**
>
> Dear Reviewer Ge5Z,
>
> Thanks for your positive review and thoughtful comments. We have revised our paper according to your comments. Please find our response to your questions below.
>
> - Why is soundness needed? Why look at the wrong label?
>
> > Please see our general response about the motivation for considering  soundness.
>
> - Lack of comparison of other methods with other metrics?
>
> > We do evaluate the methods on other intrinsic metrics as well, including the Deletion, Insertion and Saliency metrics proposed in prior work. Details can be found in the first paragraph of Section 6.1 and results in Table 1. Tables 5, 7, 9, 10 in the appendix additionally contain more comparison on various datasets.
>
> - Back-up plan of running other methods?
>
> > We think you are referring to the backup for evaluation when the saliency method does not “make its best efforts” to find masking explanations for incorrect labels. In practice, many saliency methods take in a label, and hence can compute masking explanations for incorrect labels just like the correct label. If they do not take in a label or ignore the input label, in our current experiments, we still take whatever saliency explanations they output (i.e., do not use some default method for incorrect labels), which gives them some advantage for the consistency score metric and the soundness metrics. However, our method still achieves the best consistency score and soundness.
>
> - Referring to the supplement too often
>
> > Thank you for the constructive criticism. We have moved some figures and discussions from the appendix into the main paper for a smoother flow.

---

> > ### Comment · Reviewer_Ge5Z · 2021-11-29
> > **Response to updates**
> >
> > Thank you. The paper has improved. It, however, seems to me,  it may benefit from another round of revision. I would keep my rating.

---

### Author Response · Authors · 2021-11-22
**General Response**

We thank all reviewers for their detailed reviews and suggestions. We have revised our paper accordingly. Major changes are marked in red. Here we would like to address two points that concerned reviewers the most, which we have also emphasized in our revised version.

First, we would like to clarify “soundness”,  a key concept introduced here. Several reviewers asked why it is needed. The concept is motivated by the observation that the recent mask-based saliency methods are inherently different from the earlier gradient-based methods. They produce a heatmap that is interpreted as a masked image that, when fed into the net, must make it output the same label that received the *top* logit value on the full image. Our paper observes that it is unclear at a logical level why such a mask should be considered an “explanation” for the label. It can be considered an explanation only if one also verifies that a different mask could not induce the net to output a label corresponding to other logits. Figure 1 shows that this issue is indeed real: masks “justifying” the wrong labels do exist. Soundness is a quantitative measure for this issue, and we propose that evaluations of mask-based saliency methods must account for soundness, which has been missing thus far. We have added a discussion about this in the revision in Section 1. Note that soundness is tested when evaluating the quality of the saliency method, and does not affect the use of the method.

Connection to logic: In logic, *completeness* of a proof system roughly means that “every true statement can be proven” and *soundness* means that “no false statement can be proven”. A masking-based explanation can be seen as a “proof” of the statement “the net predicts label $a$ for input $x$” where $a$ is the label with the highest logit value. Soundness then requires that an analogous “proof” should not exist for the statement  “model predicts label $a'$ for input $x$” where $a'\neq a$ and soundness. One reviewer asked whether this inherently requires that the net is giving high probability to a single label (e.g. in our zebra/elephant figure on page 3 of the revised version). It does not. For instance, if two labels $a_1, a_2$  have probability $0.45$ and the other labels have negligible probability, then our completeness/soundness notion in Definition 3.2 with suitable $\alpha, \beta$ allows explanations to exist for both $a_1$ and $a_2$.

Second, we summarize the observations from our experiments (especially Table 1). Note that we do not intend to claim that our method is the best intrinsic saliency method, nor that our metric should replace previous metrics. However, we do make a case for soundness through our experiments. We show the following: (1) our simple mask learning procedure achieves competitive performance compared to existing saliency methods on existing metrics; (2) our procedure does better on our consistency score metric compared to previous methods, which suggests that consistency score has no major conflicts with previous metrics and that our method has higher soundness; (3) TV regularization and upscaling significantly aid soundness, while only slightly hurting completeness. Overall, consistency score increases with higher TV regularization and upscaling, thus showing the ability of our metrics to provide intrinsic explanations for existing “tricks” that previously only had extrinsic justifications.  We made these contributions clearer in the introduction and start of Section 6.

---

### Decision · Program_Chairs · 2022-01-20

**Decision:**

Reject

**Comment:**

This submission tackles the problem of model explainability from the perspective of masking-based saliency methods.
Several metrics are proposed for evaluating saliency methods including  a new « soundness »  concept.
Experiments using a consistency score to simultaneously evaluate completeness and soundness are provided.

Most of the reviewers were not convinced by the approach and have raised several issues.

After rebuttal and despite the interest in the introduction of the concept of « soundness »  to better explain model decision, the current proposition needs to be improved. In particular, the interest of this soundness concept does not bring out, many details of the method are not clear enough and the effectiveness of the proposed measure is still questionable. It would be interesting that the authors consider the R’s comments as the ones for additional  experiments to demonstrate the relevancy of their contribution.